# Batch-Adaptive Annotations for Causal Inference with Complex-Embedded Outcomes

## Abstract

Estimating the causal effects of an intervention on outcomes is crucial to policy and decision-making. But often, information about outcomes can be missing or subject to non-standard measurement error. It may be possible to reveal ground-truth outcome information at a cost, for example via data annotation or follow-up; but budget constraints entail that only a fraction of the dataset can be labeled. In this setting, we optimize *which data points should be sampled for outcome information* and therefore efficient average treatment effect estimation with missing data. We do so by allocating data annotation in batches. We extend to settings where outcomes may be recorded in unstructured data that can be annotated at a cost, such as text or images, for example, in healthcare or social services. Our motivating application is a collaboration with a street outreach provider with millions of case notes, where it is possible to expertly label some, but not all, ground-truth outcomes. We demonstrate how expert labels and noisy imputed labels can be combined into a doubly robust causal estimator. We run experiments on simulated data and two real-world datasets, including one on street outreach interventions in homelessness services, to show the versatility of our proposed method.

## 1 Introduction

Evaluating causal effects of a treatment or policy intervention is a challenging problem in its own right, but an added layer of complexity comes when there is missing data. In this paper, we consider a setting of observational causal inference with missing outcomes, where it is possible to obtain information about ground-truth outcomes at a cost, via expert annotation or follow-up. Recent tools in machine learning can label outcomes, but for inferential goals, this can lead to error-prone and biased outputs. With a small budget, one can obtain valid causal effects on a small subsample without using additional contextual information or imputation, but this can be high-variance. We build on doubly-robust causal inference with missing outcomes to determine where to sample additional outcome annotations to minimize the asymptotic variance of treatment effect estimation.

Our methodology is motivated by a collaboration with a nonprofit to evaluate the impact of street outreach on housing outcomes, where rich information about outcomes of outreach are embedded in case notes written by outreach workers. Street outreach is an intensive intervention; caseworkers canvass for and build relationships with homeless clients and write case notes after each interaction. These notes are a noisy view on the ground-truth of what happens during the open-ended process of outreach. Was a client progressing on their housing application or their goals, or were they facing other barriers? In our experience, outreach workers can extract structured information, from the unstructured text of case notes. They can provide context and recognize important milestones. But it is simply impossible for under-resourced expert outreach workers to label millions of case notes. While modern natural language processing tools can facilitate annotation at scale, they are often

inaccurate. *Given an annotation budget constraint, how can we strategically collect ground truth data, such as by assigning expert annotation, while leveraging additional data sources or weaker annotation to optimize causal effect estimation?* In this paper, we develop general methodology for optimizing data annotation and we validate our methodological innovations using outcomes with plausible ground-truth information on housing placement.

This problem is not unique to the social work domain and can generally apply to cases of measurement error with misaligned modalities (such as text or images), where it is possible to query the ground truth directly for some portion of the data at a cost. In some settings, we can query other data sources for ground-truth labels directly, while in other settings, outcomes may be recorded in complex information such as text or images. However, due to dimensionality issues, these cannot be directly substituted for ground-truth outcomes $Y$. Weaker imputation of auxiliary information is feasible at scale, but second-best due to inaccuracies. For example, when an outcome variable, wages, is only observed from self-reported working individuals, surveyors could conduct follow-up interviews with participants to obtain wage data, but this can be expensive. Noisy measures from the same dataset (such as last year's wages) or transporting prediction models from national wage databases can be predictive. Such trade-offs between expert annotation and scalable, weaker imputation are pervasive in data-intensive machine learning, for example as in the recent "LLM-as-a-judge" framework [56].

This study makes the following contributions: we propose a two-stage batch-adaptive algorithm for efficient ATE estimation from complex embedded outcomes. We derive the expert labeling probability that minimizes the asymptotic variance of an orthogonal estimator [4]. We design a two-stage adaptive annotation procedure. The first stage estimates nuisance functions for the asymptotic variance on the fully observed data. We use the estimates and functions from the first stage to estimate the optimal labeling probabilities in the second stage. The final proposed estimator combines the model-annotated labels and the expert labels in a doubly robust estimator for the ATE. We show that this two-stage design achieves the optimal asymptotic variance with weaker double-machine learning requirements on nuisance function estimates. We leverage our closed-form characterizations to provide insights on how to improve downstream treatment-effect estimation. We validate and show improvements upon random sampling on semi-synthetic and real-world datasets from retail and street outreach.

## 2   Related work

Our model is closest to optimizing a validation set for causal inference with missing outcomes, which can be broadly useful for causal inference with non-standard measurement error. Typical distributional conditions for non-standard measurement error [38] are generally inapplicable to text or images, our motivating application. The most related work is that of [20, 58], which leverages the fact that sampling probabilities for data annotation are known to obtain doubly-robust estimation via causal inference. These works generally address non-causal estimands such as mean estimation and M-estimation (therefore without discussion of treatment effect estimation). Our work follows a key approach in adaptive experimental design of optimizing the semiparametric efficiency lower bound, whether via batch or full adaptivity. Hahn et al. [24] studied a two-stage procedure for estimating the ATE with a proportional asymptotic, and show asymptotic equivalence of their batched adaptive estimator to the optimal asymptotic variance. [36] and [46] also considers a double machine learning version of [24], though our estimator is different and we further characterize the closed-form optimal sampling probabilities, yielding additional analysis. Other variants in the same framework include Uehara et al. [46] which optimizes treatment choice or Cook et al. [15] which provides anytime-valid inference for full adaptivity. Armstrong [1] proves the semiparametric efficiency lower bound cannot be beat in general by adaptive designs; so this algorithmic paradigm is the right fit for our goals of efficient statistical inference. Crucially, all these other papers focus on allocating treatments, while we allocate the probability of revealing the outcome for a datapoint (i.e, via expert annotation); this gives us a different optimization problem and different estimation challenges.

Regarding the use of auxiliary information in causal inference, many recent works have studied the use of surrogate or proxy information. Although our context $\tilde{Y}$ aligns with colloquial notions of surrogates or proxies, recent advances in surrogate and proxy methods refer to specific models that differ from our direct measurement/costly observation setting [3, 32, 20]. Surrogates often estimate an outcome that is impossible to measure at the time of analysis [3], such as long-term effects; while we can obtain ground truth outcomes from expert data annotators feasibly but under a binding budget constraint. We do leverage that we can design the sampling probabilities of

outcome observations (ground-truth annotation) or missingness for doubly-robust estimation, like some methods in the surrogate literature or data combination [53, 32]. But we treat the underlying setting as a single unconfounded dataset with missingness. The different setting of proximal causal inference [44, 16] seeks proxy outcomes/treatments that are informative of unobserved confounders; we assume unconfoundedness holds. Recently, [9] study the "design-based supervised learning" perspective of [20] specifically for proxies for unobserved confounding.

Many exciting recent works study adaptive experimentation under different desiderata, such as full adaptivity, in-sample decision regret or finite-sample, non-asymptotic guarantees [21, 54, 15]. Such designs are closely related to covariate-adaptive-randomization; the recent work of [41] studies delayed outcomes. These desiderata are less relevant to our specific setting of data annotation: it's easier to leverage human annotators with batch annotation rather than full adaptivity, and in-sample regret is less meaningful for data annotation than for treatment allocation. Technical tools from these other works could be applied to our setting as well.

# 3 Problem setup

Our problem setting is causal inference with missing outcomes. We discuss extensions to a setting where outcomes are measured in a high-dimensional contextual variable $\tilde{Y}$, such as images or text.

In both cases, we assume the ground-truth data-generating process follows that of standard causal inference. A data instance $(X, Z, Y(Z))$, includes covariates $X \in \mathcal{X}$, a binary treatment $Z \in \{0, 1\}$, and potential outcomes $Y(Z)$ in the Neyman-Rubin potential outcome framework. We only observe $Y(Z)$ for the realized treatment assignment $Z$ and assume the usual stable unit value treatment assumption (SUTVA). *If* the ground-truth data were observed, we would have a standard causal inference task at hand, so the key challenge is its *missingness*. We let $R \in \{0, 1\}$ denote the presence ($R = 1$) or absence ($R = 0$) of the outcome $Y$. Therefore, our observational dataset for estimation is $(X, Z, R, RY)$, i.e. with missing outcomes. For causal identification, we generally proceed under the following assumptions:

**Assumption 1** (Treatment ignorability [25, 27, 35]). $Y(Z) \perp\!\!\!\perp Z \mid X$.

**Assumption 2** (R-ignorability [37, 4]). $R \perp\!\!\!\perp Y(Z) \mid Z, X$

Assumption 1, or unconfoundedness, posits that the observed covariates are fully informative of treatment. It is generally untestable but robust estimation is possible in its absence, e.g. via sensitivity analysis and partial identification [55, 34]. On the other hand, Assumption 2 is true by design as long as the full corpus of datapoints needing annotation is available from the outset, since we choose what datapoints are annotated for ground-truth labels based on $(Z, X)$ alone.

Although one approach is completely random sampling, we are particularly concerned with *how can we select datapoints for expert annotation for optimal estimation*? We assume the budget is limited for data annotation, but we have control over the missingness mechanism, i.e. assigning data for expert annotation. Define the propensity score and annotation (outcome observation) probability:

$$e_z(X) := P(Z = z|X) \text{ (propensity score), and } \pi(Z, X) := P(R = 1|Z, X) \text{ (annotation probability).}$$

We assume positivity/overlap; that we observe treatment and outcome with nonzero probability.

**Assumption 3** (Treatment and annotation positivity [25, 27, 35]). $\epsilon < \pi(z, X) \leq 1, z \in \{0, 1\}$ and

$1/\nu < e_1(X) < 1 - 1/\nu, \nu > 0$

We define the outcome model, which is identified on the $R = 1$ data by Assumption 2, and the conditional variance:

$$\mu_z(X) := \mathbb{E}[Y \mid Z = z, X] \underset{asn.2}{=} \mathbb{E}[Y \mid Z = z, R = 1, X]$$
$$\sigma_z^2(X) := \mathbb{E}[(Y - \mu_z(X))^2 \mid Z = z, X = x].$$

**Batch allocation setup.** We consider a two-batch adaptive protocol, where $n$ iid observations are randomly split into two batches. We consider a proportional asymptotic regime where the budget and size of first batch $n_1$ are fixed proportions $\kappa \in (0, 1)$ of the dataset size.

**Assumption 4** (Proportional asymptotic [24, 36]). $\lim_{n\to\infty} \frac{n_1}{n} = \kappa$.

In the first batch, we randomly assign annotations according to a small but asymptotically nontrivial fraction of the budgets. In the first batch, outcomes are realized and observed, and the nuisance models $(\hat{\mu}_z(x), \hat{e}_z(x), \hat{\sigma}_z^2(x))$ are trained on the observed data. We solve for optimal annotation probabilities $\pi*$ and sample data in the second batch so that the mixture distribution over outcome observations achieves $\pi*$. We combine the results from both batches and use the data for ATE estimation, which we describe in the next section.

**Extension to Missing Outcomes with Context.** We extend our missing outcome framework to cases where we have additional contextual measurements of outcomes. In this setting, our observed data includes $\tilde{Y}$, a widely available "*complex-embedded outcome*", i.e. a ground-truth outcome embedded in more complex information, such as images, text, etc. Though $\tilde{Y}$ is observable for every datapoint, it is not usable for direct estimation. We assume causal effects operate through a latent true outcome $Y$, of which $\tilde{Y}$ is a complex observation (but not deterministic function) thereof. We assume an exclusion restriction that the direct effect of treatment passes through the latent outcomes only[25, 49].

**Assumption 5** (Complex embedded outcomes: exclusion restriction[42, 28])**.**

$$\tilde{Y}(Z) = g(Y(Z), X) + \epsilon, \ \epsilon \neq 0 \ a.s.; \text{ and } Z \perp \tilde{Y} \mid X, Y(Z)$$

We don't make distributional assumptions on the measurement error mechanism, appealing instead to data annotation/validation measurement which we assume reveals the ground-truth outcome $Y$, although ultimately predicting $Y$ from $\tilde{Y}$ just needs to be consistently estimable, discussed later on.

This assumption asserts that treatment assignment does not affect $\tilde{Y}$ beyond affecting latent outcomes $Y$. For example, in a medical setting, it holds if treatment affects underlying biological phenomena, e.g. makes a tumor smaller, these phenomena are recorded via clinical notes or raw pixel images, and treatment doesn't change textual or visual expression. It assures that predicting latent outcomes $Y$ from $\tilde{Y}$ does not introduce collider bias, and is testable, i.e. after the first batch of data.

In this setting, we allow the outcome model to depend on the complex embedded $\tilde{Y}$, and denote $\mu_Z(X, \tilde{Y}) := \mathbb{E}[Y|Z, X, \tilde{Y}]$. There are a few variants of estimating this from data. We denote an ML-prediction based on $\tilde{Y}$ (with $X$ covariates and treatment information) as $f_z(X, \tilde{Y})$; for example zero-shot prediction using an LLM or pretrained model. Variants include calibrating zero-shot predictions to ground-truth $\mathbb{E}[Y \mid Z, R = 1, f_z(X, \tilde{Y})]$, predicting $Y$ and including ML predictions as a covariate alongside $X$, or various ensembling combinations thereof. This last approach is suggested in Egami et al. [20]. Later on, we find that in practice, choosing the outcome model that reduces the mean squared error leads to better numerical results.

# 4 Method

This section outlines our proposed methodology. We first recap the AIPW estimator for the missing outcomes case and provide the lower bound for the asymptotic variance. Then we consider a global budget optimization problem and solve for the optimal $\pi^*(z, x)$. (In Appendix F we discuss a very similar case of treatment-specific budget). We describe feasible estimation of the ATE by the AIPW estimator (with missing outcomes).

**Recap: Optimal asymptotic variance for the ATE with missing outcomes.** Our target parameter of interest is the ATE of a binary treatment vector $Z$ on an outcome $Y$.

$$\tau = \mathbb{E}[Y(1) - Y(0)].$$

Bia et al. [4] derives a double-machine learning estimator for ATE estimation with missing outcomes:

$$\mathbb{E}[Y(z)] = \mathbb{E}[\psi_z], \text{where } \psi_z = \frac{\mathbb{I}[Z = z]R(Y - \mu_z(X))}{e_z(X)\pi(z, X)} + \mu_z(X), \text{ and } \tau_{AIPW} = \mathbb{E}[\psi_1 - \psi_0].$$

The outcome model $\mu_z(X)$ is estimated on data with observed outcomes, since SUTVA and assumption 2 give that $\mathbb{E}[Y(z)|X] = \mathbb{E}[Y|Z = z, X] = \mathbb{E}[Y|Z = z, R = 1, X]$.

The focus of our work is to optimize the semiparametric efficient asymptotic variance (proven in [4]), which is closely related to the ATE of [23].

**Proposition 1.** *The asymptotic variance (AVar) is:*

$$\text{AVar} = \text{Var}[\mu_1(X) - \mu_0(X)] + \sum_{z \in \{0,1\}} \mathbb{E}\left[\frac{\sigma_z^2(X)}{e_z(X)\pi(z,X)}\right]$$

The first term is independent of $\pi$; we focus on optimizing the second term with respect to $\pi$.

*Remark* 1. We state the results for the base model, though they extend directly for the case with contexts. With contexts, by marginalizing over $\tilde{Y}$, the analogous expressions use the estimators $\hat{\mu}_z(X, \tilde{Y})$ instead of $\hat{\mu}_z(X)$ whereas $\hat{\sigma}_z^2(X)$ stays the same (sampling probabilities depend only on $(Z, X)$ and just correspondingly marginalizes over $\tilde{Y}$, $\hat{\sigma}_z^2(X) = \mathbb{E}[(Y - \hat{\mu}_z(X, \tilde{Y}))^2 \mid Z = z, X = x]$. In the setting with noisy measurements $\tilde{Y}$, under the exclusion restriction Assumption 5, the mean potential outcome is identified by regression adjustment: $\mathbb{E}[Y(z)] = \mathbb{E}[\mathbb{E}[Y|Z = z, R = 1, X, \tilde{Y}]] = \mathbb{E}[\mathbb{E}[Y|Z = z, R = 1, X]]$.

**Characterizing the optimal** $\pi^*(z, x)$**.** We first characterize the population optimal sampling probabilities $\pi^*(z, x)$, assuming the nuisance functions are known. We optimize the asymptotic variance over $\pi$ under a sampling budget. We consider a global budget constraint $B \in [0, 1]$ over all annotations. The setting is meaningful when the budget binds, $B \ll 1$, which is still practically relevant.

$$\min_{0 < \pi(z,x) \leq 1, \forall z, x} \sum_{z \in \{0,1\}} \mathbb{E}\left[\frac{\sigma_z^2(X)}{e_z(X)\pi(z,X)}\right] \text{ s.t. } \mathbb{E}[\pi(Z, X)] \leq B \qquad \text{(OPT (global budget))}$$

Note that in the global budget constraint, $\mathbb{E}[\pi(Z, X)] = \mathbb{E}[\pi(1, X)\mathbb{I}[Z = 1] + \pi(0, X)\mathbb{I}[Z = 0]]$. We can characterize the solution as follows.

**Theorem 1.** *The solution to the global budget problem is:*

$$\pi^*(z, X) = \frac{\sqrt{\sigma_z^2(X)}}{e_z(X)} B \left(\mathbb{E}\left[\sqrt{\sigma_1^2(X)} + \sqrt{\sigma_0^2(X)}\right]\right)^{-1}$$

Note that sampling probabilities increase in the conditional variance/uncertainty of the model, $\sigma^2(X)$, and the inverse propensity score. Characterizing the closed-form solution is useful for our analysis later on, in establishing convergence of estimation to the limiting optimal data annotation probabilities. For the proof, see Appendix G

**Feasible two-batch adaptive design and estimator.** Our characterizations above assume knowledge of the true $\sigma_z^2(x)$ and the propensity scores $e_z(x)$. Since these need to be estimated, we leverage the double machine learning (DML) framework and conduct a feasible two-batch adaptive design [10, 4]. Cross-fitting with iid data [10] splits the data, estimates nuisance functions on one fold, and evaluates the estimator on a datapoint leveraging nuisance functions from another fold of data.

We leverage a variant [36] that introduces folds within each batch of data. Figure 3 summarizes the cross-fitting approach; we leave details to the appendix. First, we split the observations in each batch $t = 1, 2$ into $K$ folds (e.g. $K = 5$). Let $\mathcal{I}_k$ denote the set of batch and observation indices $(t, i)$ assigned to fold $k$ and batch $t$. Then within each fold, we estimate nuisance models on observations in batch 1. We use cross-fitting to optimize the sampling probabilities, i.e. $\pi^{*,(-k)}$ optimizes asymptotic variance with out-of-fold nuisances $e^{(-k)}$. Finally we adaptively assign annotation probabilities in batch 2. This ensures independence, that is the nuisance models only depend on observations in the previous batch from the same fold. The adaptive procedure with CSBAE cross-fitting procedure to estimate $\tau_{AIPW}$ is summarized in Algorithm 1. See Algorithm 2 for a full description.

Therefore the cross-fitted feasible estimator takes the form $\hat{\tau}_{AIPW} = \frac{1}{n} \sum_{t=1}^{2} \sum_{k=1}^{K} \sum_{(t,i) \in \mathcal{I}_k} \hat{\psi}_{1,i} - \hat{\psi}_{0,i}$ where

$$\hat{\psi}_{z,i} = \frac{\mathbb{I}[Z_i = z]R_i(Y_i - \hat{\mu}_z^{(-k)}(X_i))}{\hat{e}_z^{(-k)}(X_i)\hat{\pi}^{(-k)}(z, X_i)} + \hat{\mu}_z^{(-k)}(X_i). \tag{1}$$

## 5 Analysis

In this section, we provide a central limit theorem (CLT) for the setting where annotation probabilities are assigned adaptively and nuisance parameters must be estimated. We provide some insights to improve estimation as well as an extension to settings with continuous treatments.

---

**Algorithm 1** Batch Adaptive Causal Estimation With Complex Embedded Outcomes

---

**Input:** Data $\mathcal{D} = \{(X_i, Z_i)\}_{i=1}^n$, sampling budget $B \in [0,1]$

Step 1: Partition $\mathcal{D}$ into 2 batches and K folds $\mathcal{D}_1^{(k)}, \mathcal{D}_2^{(k)}$ for $k = 1, \ldots, K$

Step 2: On Batch 1, sample $R_1 \sim Bern(B)$. Estimate nuisances within each k-fold $\hat{\mu}_z^{(k)}(X, \tilde{Y})$, $\hat{\sigma}_z^{2(k)}(X)$, and $\hat{e}_z^{(k)}$.

Step 3: On Batch 2, folds $k = 1, \ldots, K$, obtain $\pi^*$ by optimizing eq. (OPT (global budget)), plugging in nuisance estimates. Solve for $\hat{\pi}_2^{(k)}(X_i) = \frac{1}{1-\kappa}(\pi^*(X_i) - \kappa\pi_1)$

Step 4: On Batch 2, sample $R_2 \sim Bern(\hat{\pi}_2^{(k)}(Xi))$ and obtain outcomes.

Step 5: Pool data across batches and estimate ATE with AIPW estimator in eq. (1) (or eq. (RZ-plug-in.), or balancing weights) and out of fold nuisances.

---

Denote $\|\cdot\|_2 = (\mathbb{E}[(\cdot)^2])^{1/2}$. The following assumptions can also be found in [36, 10, 49, 46, 4].

**Assumption 6** (Consistent estimation and boundedness). Assume bounded second moments of outcomes and errors, $\|Y(z)\|_2 \leq C_1$, $\|\mu_z(X)\|_2 \leq C_2$, $\|(Y - \mu_z(X))\|_2^2 \leq 4B_{\sigma^2}$, $\forall z$; and consistent estimation $\mathbb{E}[(\mu_z(X) - \hat{\mu}_z(X))^2] \leq K_\mu n^{-r_\mu}$ for some constants $C_1, C_2, B_{\sigma^2}, K_\mu, r_\mu \geq 0$.

**Assumption 7** (Product error rates [4]). For nuisance functions, assume the products of their mean-square convergence rates vanish faster than $n^{-1/2}$: (i) $\sqrt{n}\|\hat{\mu}_z(X) - \mu_z(X)\|_2 \times \|\hat{\pi}(z, X) - \pi(z, X)\|_2 \xrightarrow{p} 0$; (ii) $\sqrt{n}\|\hat{\mu}_z(X) - \mu_z(X)\|_2 \times \|\hat{e}_z(X) - e_z(X)\|_2 \xrightarrow{p} 0$.

**Assumption 8** (VC dimension for nuisance estimation[2]). The nuisance estimation of $e_z$, and $\sigma_z^2$ occurs over function classes with finite VC-dimension.

**Assumption 9** (Sufficiently weak dependence across batches).

$$\sqrt{\frac{1}{n_{t,k}} \sum_{i:(t,i)\in\mathcal{I}_k} \left\| \mathbb{E}\left[ \hat{\psi}_i(R; \hat{e}^{(-k)}, \hat{\pi}^{(-k)}, \hat{\mu}^{(-k)}) - \psi_i(R; e^{(-k)}, \pi^{(-k)}, \mu^{(-k)}) \mid \mathcal{I}^{(-k)}, X_i \right] \right\|^2} = o_p(n^{-\frac{1}{4}})$$

**Theorem 2.** *Given Assumptions 1 to 3, suppose that we construct the feasible estimator $\hat{\tau}_{AIPW}$ (Equation (1)) using the CSBAE crossfitting procedure in Figure 3 with estimators satisfying Assumptions 6 and 7 (consistency and product error rates). Then*

$$\sqrt{n}(\hat{\tau}_{AIPW} - \tau) \Rightarrow \mathcal{N}(0, V_{AIPW}),$$

*where $V_{AIPW} = \sum_{z\in 0,1} \mathbb{E}\left[\frac{\sigma_z^2(X)}{e_z(X)\pi^*(z,X)}\right] + \text{Var}[\mu_1(X) - \mu_0(X)]$. Here $\tau$ is the ATE.*

For the proof, see Appendix G. The main result from Theorem 2 shows that the batch adaptive design and feasible estimator has an asymptotic variance equal to the variance of the true ATE under missing outcomes and the optimal $\pi^*$. This implies that our procedure successfully minimizes the asymptotic variance bound. With this, we can also quantify the uncertainty of our treatment effect estimates by producing level-$\alpha$ confidence intervals for $\tau$ that achieve coverage with $1 - \alpha$ probability.

**Insights and improvements**

**When is our method much better than uniform sampling?** Prior works of [20, 59], though they do not study treatment effect estimation, obtain valid inference with uniform sampling (i.e. with the budget probability). When do optimized data annotation probabilities improve upon uniform sampling? To answer this, we analyze the relative efficiency.

**Corollary 1** (Relative efficiency). *The relative efficiency of estimation with optimized sampling probabilities $\pi$ vs. uniform sampling, for the same budget, is*

$$\text{RelEff} = \frac{\text{AVar } of\ estimation\ with\ \pi^*}{\text{AVar } of\ estimation\ with\ uniform\ prob.\ B} = \frac{\frac{1}{B}\left(\mathbb{E}\left[\sqrt{\sigma_1^2(X)} + \sqrt{\sigma_0^2(X)}\right]\right)^2 + \text{Var}[\tau(X)]}{\frac{1}{B}\mathbb{E}\left[\frac{\sigma_1^2(X)}{e_1(X)} + \frac{\sigma_0^2(X)}{e_0(X)}\right] + \text{Var}[\tau(X)]}$$

By construction, $\text{RelEff} \leq 1$; the smaller it is, the larger the improvement from our method. The above expression reveals our method improves further as the budget grows smaller ($B \downarrow$) or if there

246 are imbalanced propensities where $e_1(X)$ close to 0 or 1. On the other hand, improvements from our
247 method are limited for large budgets, $B \to 1$, or when variances in treated/control group are similar.

248 **Direct estimation of $(e\pi^*)^{-1}$ mitigates estimation stability.** It is well known that estimating
249 propensities and then inverting estimates can be unstable in practice. This problem is doubly-so
250 for causal inference with missing outcomes. We find many papers on adaptive treatment allocation
251 note this challenge and mix their optimized allocation probabilities with uniform in the experimental
252 sections [18, 59, 15]; just as many papers in causal inference clip the weights in practice [50].
253 Our near closed-form solution reveals that it's not necessary to estimate propensity scores for the
254 final ATE estimation on the full dataset (though it is needed to estimate $\pi^*$). At $\pi^*$, observe that[1]
255 $(e_z(x)\pi^*(z,x))^{-1} \propto \sqrt{\sigma_z^2(x)^{-1}}$ and is independent of the propensity score $e_z(x)$, so estimating it
256 directly can directly exploit its *lower* statistical complexity. In causal inference and covariate shift,
257 many methods (such as balancing weights) avoid the plug-in approach for inverse propensity methods
258 in favor of direct estimation of the inverse propensity score [45, 60, 26, 30, 31, 13, 5]. We recommend
259 estimation on the final dataset with such approaches or other types of direct estimation. For example,
260 even estimation of $P(Z = z, R = 1 \mid X)$ directly helps:

$$\psi_z(e, \pi^*) = \tfrac{\mathbb{I}[Z=z, R=1]}{P(Z=z, R=1 \mid X)}(Y - \mu_z(X)) + \mu_z(X). \qquad (\textit{RZ-plug-in.})$$

**Extension to continuous treatments.** Our analysis applies readily to other static causal inference
estimands, such as those for continuous treatments. We introduce the analogous estimator and the
optimal sampling probabilities. Let $e(z, X) = P(Z = z \mid X)$ be the generalized propensity score
and $\mu(Z, X) = \mathbb{E}[Y \mid Z, X]$. The estimator for continuous treatments replaces the indicator function
$\mathbb{I}[Z = z]$ with a local kernel function smoother localizing around $z$, $K_h(Z - z)$.[2] The following
estimator for continuous treatments with missing outcomes is a direct extension of [33, 14]:

$$\psi(e, \mu) = \mu(z, X_i) + \tfrac{K_h(Z_i - z)\mathbb{I}[R=1]}{e(z, X_i)\pi(z, X_i)}(Y_i - \mu(z, X_i)); \quad \mathbb{E}[Y(z)] = E[\psi(e, \mu)]$$

We consider the same assumptions required as in [14], standard in kernel density estimation analysis.
The optimal sampling probabilities minimize the part of the asymptotic variance of $\mathbb{E}[Y(z)]$ depending
on $\pi$, subject to a budget constraint:

$$\pi^*(z, x) \in \arg\min_{\pi(z,x)} \left\{ \mathbb{E}\left[\tfrac{\sigma^2(z, X)}{e(z, X)\pi(z, X)}\right] : \mathbb{E}[\pi(z, X)K_h(Z - z)] \le B \right\},$$

**Theorem 3.** *Define the kernel localization of the generalized propensity score $e(z, x)$ around $z$ under
the kernel function $K_h(z' - z)$: $\tilde{e}_h(z, x) = \int K_h(z' - z)e(z', x)dz'$. Then*

$$\pi^*(z, x) \propto \tfrac{\sqrt{\sigma^2(z, x)}}{e(z, x)}\sqrt{\tfrac{e(z, x)}{\tilde{e}_h(z, x)}}$$

261 The optimal sampling probabilities are quite similar, with the appropriate analogous conditional
262 variance and generalized propensity score, up to a factor $(e(z, x)/\tilde{e}_h(z, x))^{1/2}$ from the implications
263 of kernel-smoothing treatment for sampling budget. Consider a box kernel for simplicity, then
264 $\tilde{e}_h(z, x)$ is the average of $e(z, x)$ over the interval $e(z - h, x), e(z + h, x)$.

# 6 Experiments

266 We evaluate our batch adaptive allocation protocol on synthetic and ground-truthed real-world datasets.
267 We compare our method to a baseline (uniform random sampling in the same doubly-robust estimator)
268 and a skyline (running the estimator on the complete dataset, which is generally infeasible). The
269 baseline does not use our adaptively learned $\hat{\pi}(z, x)$, but instead uses uniform random sampling at
270 different budget values. The skyline that we compare against is the standard AIPW estimator with
271 fully observed outcomes, that is when the budget equals 1 or $R = 1$ for all data points.

272 **A note on active learning.** We also run pool-based active learning baselines. However, there are key
273 differences that lead to poor performance of these baselines. In our setting, random sampling is the

---

[1]This depends on some joint properties of $\kappa, p_1$, whether it is feasible to find second-stage batch sampling
probabilities $\pi_2$ so that $\kappa p_1 + (1 - \kappa)\pi_2(x) = \pi^*(x)$

[2]The kernel function $K_h(u)$, used in kernel density estimation, satisfies $\int_{-\infty}^{\infty} K(u)du = 1$ (normalizes to
a probability density) and $K(-u) = K(u)$, for all $u$ (symmetry), such as the Gaussian kernel with $K(u) = (2\pi)^{-\frac{1}{2}}e^{-u^2/2}$, or uniform $K(u) = 1/2\mathbb{I}[|u| \le 1]$. We generally consider $K_h(u) = h^{-d}K(u/h)$.

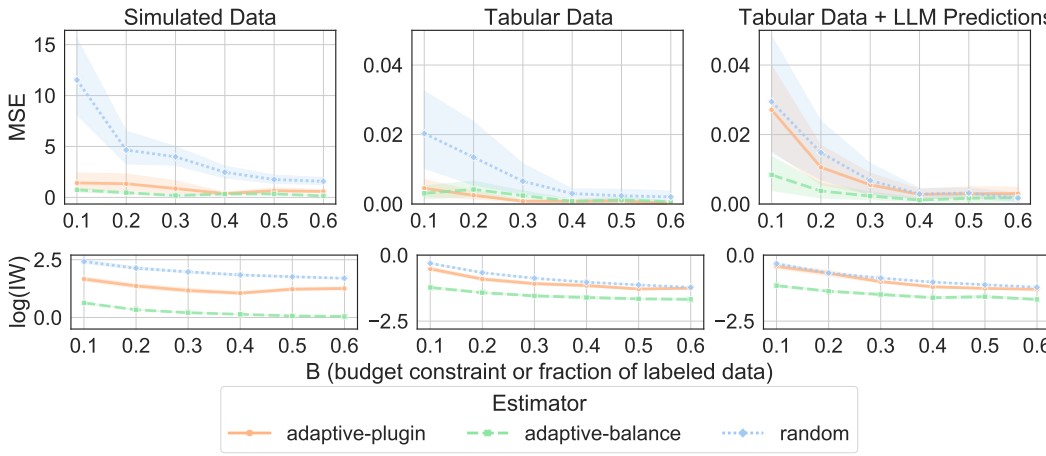

Figure 1: **Synthetic and Semi-Synthetic (Retail Hero) Data.** Mean squared error (top) and $95\%$ confidence interval width on the log scale (bottom) averaged over 20 and 100 (for simulated data) trials across budget percentages of the data. Left and Center: Random forest prediction on tabular data. Right: including LLM predictions on text and serialized features.

strong baseline, because the sampling probabilities are much simpler. Any other sampling strategy in our 2-stage framework with AIPW performs suboptimally (since we've proved that ours is optimal). Additionally, the baselines used in related papers are either random sampling or the exclusion of model-based predictions (i.e. $\hat{\mu}$ or $\tilde{Y}$). However, because our task is inherently causal, our AIPW estimator relies on $\mu$. We provide more details about active learning and baseline experiment results that we run in Appendix I.7

**Retail Hero Data.** We study a semi-synthetic dataset, RetailHero [52], augmented by Dhawan et al. [17] to include outcomes recorded in text. The dataset contains background customer information $X$, treatment $Z$ as a text message ad sent to the customer, and outcomes $Y$ of whether the customer made a purchase or not. Dhawan et al. [17] sampled datapoints according to an artificial propensity score and generated text from the binary outcomes prompting LLMs to generate social media posts following personas (given covariates) (details in Appendix I). These text posts are $\tilde{Y}$. The goal is to estimate the causal effect of SMS communication on purchase. This is an example of our contextual setting, where plentiful social media posts can offer insights into customer behaviors but companies may only be able to allocate a fixed amount of resources for ground-truth validation.

We implement our proposed methods using 1) random forest models to estimate the outcome model $\hat{\mu} = \mathbb{E}[Y|X]$ on tabular data only or 2) sampling from a set of five LLM predictions of purchase from social media posts $\tilde{Y}$ and then using them as predictors in a random forest to estimate $f(X, \tilde{Y})$ (We run the LLM predictions offline in batch to save cost and time). Then we estimate the outcome model $\hat{\mu}$ by ensembling, taking a weighted average between $\mathbb{E}[Y|X]$ (random forest) and $\mathbb{E}[Y|X, f(X, \tilde{Y})]$ (support vector machine), choosing the best models and weights to minimize the MSE of predicting $Y$ on $20\%$ of the full data. We average the results over 20 random data splits. We compute the AIPW estimator on all available data as a stand-in for ground-truth. (The dataset was too small for a separate held-out validation set). We have further experiments with simulated data to validate these results (more details on the data-generating process are given in Appendix I.)

Figure 1 shows the performance of our adaptive estimator either with 1) a direct estimation of $(e\pi^*)^{-1}$ using logistic regression that we plug-in (following Equation ($RZ$-plug-in.)) or 2) a random forest-based estimator of $(e\pi^*)^{-1}$ extracted from ForestRiesz [11], a random forest-based method to learn balancing weights, compared to a uniform baseline. Across different values of the budget, $B$, our batch adaptive procedure reduces the MSE by almost double and reduces the confidence interval width by almost one-unit in the interval width on the log scale. In Figure 10, we see the impact of our approach most clearly when we compute the percentage of the budget saved to reach the same interval width. We observe a minimum budget saved of $10\%$ with the adaptive plug-in estimator and $45\%$ with the adaptive balance estimator on tabular data. The LLM prediction we generate is based on simple zero-shot learning and direct serialization of the tabular data; further fine-tuning

309 could improve performance. Nonetheless, our method can provide robust valid guardrails around
310 these black-box predictions.

311 **Street Outreach Data.** Next, we demonstrate our method on street outreach casenote data collected
312 by a partnering nonprofit providing homelessness services. This analysis, which uses proprietary
313 sensitive data, was approved by the Institutional Review Boards at [blinded for review].

314 The covariate data $X$ consists of baseline characteristics on each client as tabular data (left, Figure 2),
315 such as the number of previous outreach engagements, and (right, Figure 2) LLM generated summaries
316 of case notes recorded before treatment. We construct the cohort in our dataset to include clients who
317 are seen consistently at least once per month from 2019-2021. The binary treatment $Z$ was based on
318 the number of outreach engagements within the first 6 months of 2019. Clients with 1-2 engagements
319 were assigned $Z = 0$ (131 clients), and those with 3-15 where assigned $Z = 1$ (355 clients). The
320 outcome $Y$ is the highest housing placement reached by 2021. Our final data set contained 471
321 clients. More information on the data can be found in Appendix I. We seek to estimate the causal
322 effect of street outreach on housing placement. We use housing placement as an illustrative example
323 because it is well-recorded ground truth data in our dataset. However, it could also be plausibly
324 missing, in which case nonprofits have to decide how to expend their limited resources to obtain more
325 information (i.e., caseworker follow-up calls or analyzing more recent casenotes $\tilde{Y}$).

326 Similar to Retail Hero, we demonstrate the utility of our approach by using a random forest model to
327 estimate the outcome model on tabular data alone, $\hat{\mu} = \mathbb{E}[Y|X]$, and we incorporate LLM predictions
328 $f(X)$ by including them as predictors in a random forest model to get $\hat{\mu} = \mathbb{E}[Y|X, f(X)]$.

329 In Figure 2 we see that overall our adaptive approach shows improvements over uniform random
330 sampling. The MSE is doubled when going from both adaptive estimators to random sampling in
331 the tabular data setting and tripled with LLM predictions from the adaptive estimator with balancing
332 weights to random sampling. In Figure 11, we see that we can save between $43 - 75\%$ of the
333 budget using the plugin-in estimator on tabular data alone and by incorporating LLM predictions,
334 and between $53 - 91\%$ using the balance estimator over the random sampling baseline.

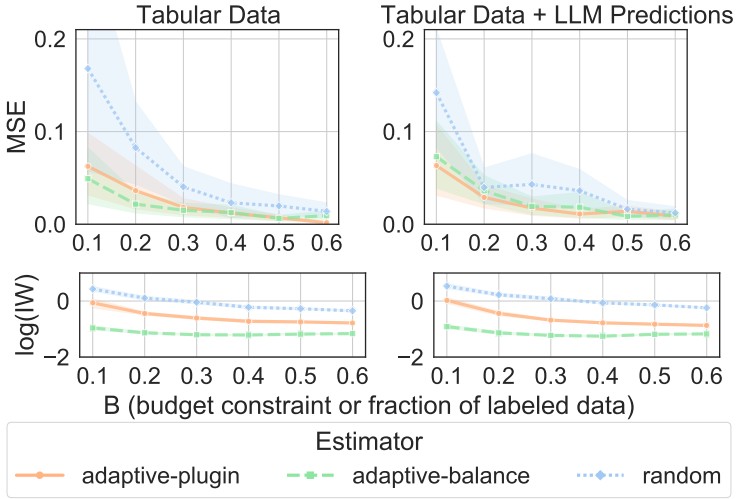

Figure 2: **Street Outreach Data.** Mean squared error and $95\%$ confidence interval width averaged
over 20 trials across budget percentages of the data. This plot makes use of tabular data and the
best-performing random forest outcome model (left) and text-encoded outcomes using LLMs (right).

335 **Conclusion, limitations, and future work.** We have introduced a batch-adaptive annotation
336 procedure and estimators that provides a framework for efficient data labeling and incorporating
337 complex embedded outcomes into causal estimation. This work is not without limitations. We assume
338 that annotations reveal ground truth, but there could be disagreement between expert annotators.
339 Additionally, LLMs are still a black box and our theory requires them to be consistent to satisfy
340 product error rate assumptions. In future work, we plan to explore other causal estimators.

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

 # A   Impact Statement

Our work deals with sensitive information about a vulnerable community so care must be taken when
deploying our methods. The case notes are redacted by the organization, and any sensitive information
is removed from the notes. Furthermore, we use local LLMs accessed through a HIPAA-compliant
fire-walled cloud instance to mitigate ensure the privacy of clients. We work in collaboration with a
nonprofit to ensure that the necessary guardrails are in place and that their data is used responsibly
and in line with their mission.

# B   Notation

| | |
|---|---|
| $Y_i$ | Ground truth outcomes, observed when label is provided by experts |
| $\tilde{Y}_i$ | Complex embedded outcomes, such as raw text |
| $X_i$ | Covariates included in estimation |
| $Z_i$ | Treatment assignment indicator |
| $R_i$ | Missingness indicator, indicates whether $i$ is expertly labeled |
| $e_z(X_i)$ | Propensity score, probability of being assigned treatment $Z = z$ |
| $\pi(Z_i, X_i)$ | Annotation probability, probability of sampling unit $i$ for expert annotation |
| $f(\tilde{Y}_i)$ | Estimated function of complex embedded outcomes, e.g. zero-shot LLM prediction from raw text |
| $\mu_z(X_i, f(\tilde{Y}_i))$ | Estimated model predicting $Y$ as function of $f(\tilde{Y})$ alone or $(X, f(\tilde{Y}))$ |

# C   Additional discussion on related work

**Additional discussion on surrogate estimation**   In much of the surrogate literature, surrogates
measure an outcome that is impossible to measure at the time of analysis. The canonical example
in [3] studies the long-term intervention effects of job training on lifetime earnings, by using only
short-term outcomes (surrogates) such as yearly earnings. In this regime, the ground truth cannot
be obtained at the time of analysis. In this paper, we focus a different regime where obtaining the
ground truth from expert data annotators is feasible but budget-binding.

**Additional discussion on more adaptive allocation methods beyond batch.**   We outline how our
approach is a good fit for our motivating data annotation setting. Full-adaptivity is less relevant in our
setting with ground-truth annotation from human experts, due to distributed-computing-type issues
with random times of annotation completion. But standard tools such as the martingale CLT can be
applied to extend our theoretical results to full adaptivity. Additionally, many recent works primarily
focus on the different problem of treatment allocation for ATE estimation. In-sample regret is less
relevant for our setting of data annotation, which is a pure-exploration problem.

**Optimizing asymptotic variance of the ATE vs. active learning.**   An extensive literature in
machine learning studies where to sample data to improve machine learning predictors, in the subfield
of active learning. The biggest difference is that we target functional estimation, aka improving
estimation and inference on the average treatment effect, rather than improving estimation of the
black-box nuisance predictors, so our approach is complementary to other approaches for active
learning. Approaches for active learning with nonparametric regression include Zhu and Nowak
[57], Chaudhuri et al. [8]. Active learning generally requires additional structural conditions, such
as margin or low-noise conditions, in order to show improvements. Our work highlights optimality
leveraging the structure of our final treatment effect inferential goal.

**Relationship to causal inference and NLP**   There is a large and rapidly growing literature on
causal inference with text data [19, 43, 47]. Throughout, we have deliberately used the terminology
of measurement error to characterize our approach: that text measures outcomes of interest. [17] also
adopt this stance towards text and note that it differs from prior works on causal inference and NLP,
which focuses on questions of substantive interest related to the text itself.

Although we can define a potential outcome $\tilde{Y}(Z)$, we are generally uninterested in causal inference
in the ambient high-dimensional space of $\tilde{Y}(Z)$ itself - corresponding to, in our examples, the

effect of the presence of a tumor on the pixel image, the effect of street outreach on the linguistic characteristics of casenotes written for documentation, etc — $\tilde{Y}(Z)$ is relevant to causal estimation insofar as it is informative of latent outcomes $Y(Z)$.

This is consistent with viewing certain types of NLP tasks as "anti-causal learning" [39], wherein outcomes cause measurements thereof, in analogy to anti-causal learning in supervised classification where a label of "cat" or "dog" causes the classification covariates (e.g. image) [29]. Analogously, we view the underlying ground-truth outcomes $Y$ as causing the measurement thereof, $\tilde{Y}$.

## D  Diagram of Cross-fitting Procedure

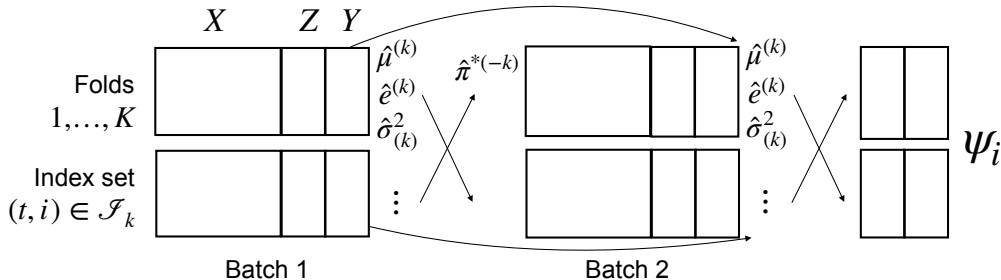

Figure 3: Illustration of cross-fitting ($K$ folds within batches)

 # E  Algorithm

---

**Algorithm 2** (Full Algorithm) Batch Adaptive Causal Estimation With Complex Embedded Outcomes

---

**Input:** Data $\mathcal{D} = \{(X_i, Z_i, Y_i, \tilde{Y}_i)\}_{i=1}^n$, sampling budget $B_z$ for $z \in \{0, 1\}$

**Output:** ATE estimator $\hat{\tau}_{AIPW}$

Partition $\mathcal{D}$ into 2 batches and K folds $\mathcal{D}_1^{(k)}, \mathcal{D}_2^{(k)}$ for $k = 1, \ldots, K$

*Batch 1:*

**for** $k = 1, \ldots, K$ **do**

    On $\mathcal{D}_1^{(k)}$: Sample $R_1 \sim \text{Bern}(\pi_1(Z, X))$, where $\pi_1(z, x) = B_z$.

    Estimate nuisance models: Where $R = 1$, estimate $\hat{\mu}_z^{(k)}$ by regressing $Y$ on $X, \tilde{Y}$, and $\hat{\sigma}_z^{2(k)}$ by

    regressing $(Y - \hat{\mu}_z)^2$ on $X$. Estimate $\hat{e}_z^{(k)}$ by regressing $Z$ on $X$.

**end for**

*Batch 2:*

**for** $k = 1, \ldots, K$ **do**

    On $\mathcal{D}_2^{(k)}$: Obtain $\pi^*$ by optimizing eq. (OPT (global budget)), plugging in $\hat{\mu}_z^{(-k)}, \hat{\sigma}_z^{2(-k)}$, and

    $\hat{e}_z^{(-k)}$.

    Solve for $\hat{\pi}_2^{(k)}(X_i) = \frac{1}{1-\kappa}(\pi^*(X_i) - \kappa\pi_1)$

    Sample $R_2 \sim \text{Bern}(\hat{\pi}_2^{(k)}(X_i))$

**end for**

Obtain $\mathcal{D}^{(k)}$ for $k = 1, \ldots, K$ by pooling across batches $\mathcal{D}_1^{(k)}$ and $\mathcal{D}_2^{(k)}$

On $\mathcal{D}^{(k)}$, re-estimate $\hat{\mu}_z^{(k)}, \hat{\sigma}_z^{2(k)}$, and $\hat{e}_z^{(k)}$ on observed outcomes $RY$ for $k = 1, \ldots, K$

On $\mathcal{D}^{(k)}$, run optimization procedure to get $\pi^{*(-k)}$ with out of fold nuisances $\hat{\mu}_z^{(-k)}, \hat{\sigma}_z^{2(-k)}$, and

$\hat{e}_z^{(-k)}$.

On full data $\mathcal{D}$, estimate ATE by using AIPW estimator in eq. (1) and out of fold nuisances $\pi^{*(-k)}$,

$\hat{\mu}_z^{(-k)}, \hat{\sigma}_z^{2(-k)}$, and $\hat{e}_z^{(-k)}$

---

 # F  Additional Results

 ## F.1  Treatment$-z$-specific budgets $B_z$

We also consider a setting with different a priori fixed budgets within each treatment group, where

$$\text{sampling budget proportion } B_z \in [0, 1]$$

is the max percentage of the treated group $Z = z$ that can be annotated. Given that we are trying to choose the $\pi$ that minimizes this variance bound, we only need to focus on the terms that depend on $\pi$ and can drop the rest. Supposing oracle knowledge of propensities and outcome models, the optimization problem, for each $z \in \{0, 1\}$ is:

$$\min_{0 < \pi(z,x) \leq 1, \forall z,x} \left\{ \mathbb{E}\left[\frac{\sigma_z^2(X)}{e_z(X)\pi(z, X)}\right] : \mathbb{E}\left[\pi(z, X) \mid Z = z\right] \leq B_z, z \in \{0, 1\} \right\} \quad \text{(z-budget)}$$

**Theorem 4.** *The solution to the within-$z$-budget problem is:*

$$\pi^*(z, X) = \frac{\sqrt{\sigma_z^2(X)/e_z^2(X)}}{\mathbb{E}\left[\sqrt{\sigma_z^2(X)/e_z^2(X)} \mid Z = z\right]} \cdot B_z$$

 .

# G   Proofs

683 *Proof of Proposition 1* . We simplify the expression for the asymptotic variance of the ATE with
684 missing outcomes to isolate the components affected by the data annotation probability.

685 First the variance of the ATE defined in terms of the efficient influence function $\psi_z$ is

$$
\begin{aligned}
\mathrm{Var}[\psi_z - \psi_{z'}] = \mathrm{Var}&\left[ \frac{Z \cdot R \cdot [Y - \mu_z(X)]}{e_z(X) \cdot \pi(z, X)} + \mu_z(X) - \frac{Z' \cdot R \cdot [Y - \mu_{z'}(X)]}{e_{z'}(X) \cdot \pi(z', X)} + \mu_{z'}(X) \right] \\
= &\underbrace{\mathrm{Var}\left[ \frac{Z \cdot R \cdot [Y - \mu_z(X)]}{e_z(X) \cdot \pi(z, X)} + \mu_z(X) \right]}_{V_1} + \underbrace{\mathrm{Var}\left[ \frac{Z' \cdot R \cdot [Y - \mu_{z'}(X)]}{e_{z'}(X) \cdot \pi(z', X)} + \mu_{z'}(X) \right]}_{V_2} \\
& \underbrace{- 2\mathrm{Cov}\left[ \frac{Z \cdot R \cdot [Y - \mu_z(X)]}{e_z(X) \cdot \pi(z, X)} + \mu_z(X), \frac{Z' \cdot R \cdot [Y - \mu_{z'}(X)]}{e_{z'}(X) \cdot \pi(z', X)} + \mu_{z'}(X) \right]}_{V_3}
\end{aligned}
$$

686 **For $V_3$:**

$$
\begin{aligned}
&2\mathrm{Cov}\left[ \frac{Z \cdot R \cdot [Y - \mu_z(X)]}{e_z(X) \cdot \pi(z, X)} + \mu_z(X), \frac{Z' \cdot R \cdot [Y - \mu_{z'}(X)]}{e_{z'}(X) \cdot \pi(z', X)} + \mu_{z'}(X) \right] \\
=&2\left[ \mathbb{E}\left[ \frac{Z \cdot R}{e_z(X) \cdot \pi(z, X)} \underbrace{[\mathbb{E}[Y|Z, R = 1, X] - \mu_z(X)]}_{=0} \right] \right. \\
&+ \left[ \mathbb{E}\left[ \mu_z(X) \cdot \frac{Z' \cdot R}{e_{z'}(X) \cdot \pi(z', X)} \underbrace{[\mathbb{E}[Y|Z', R = 1, X] - \mu_{z'}(X)]}_{=0} + \mu_{z'}(X) \right] \right] \\
&- \mathbb{E}\left[ \frac{Z \cdot R}{e_z(X) \cdot \pi(z, X)} \underbrace{[\mathbb{E}[Y|Z, R = 1, X] - \mu_z(X)]}_{=0} + \mu_z(X) \right] \\
&\left. \times \mathbb{E}\left[ \frac{Z' \cdot R}{e_{z'}(X) \cdot \pi(z', X)} \underbrace{[\mathbb{E}[Y|Z', R = 1, X] - \mu_{z'}(X)]}_{=0} + \mu_{z'}(X) \right] \right] \\
=&2\left[ \mathbb{E}[\mu_z(X) \cdot \mu_{z'}(X)] - \mathbb{E}[\mu_z(X)\mu_{z'}(X)] \right]
\end{aligned}
$$

687 **For $V_1$:**

$$
\begin{aligned}
&\mathrm{Var}\left[ \frac{Z \cdot R \cdot [Y - \mu_z(X)]}{e_z(X) \cdot \pi(z, X)} + \mu_z(X) \right] \\
=& \mathrm{Var}\left[ \frac{Z \cdot R \cdot [Y - \mu_z(X)]}{e_z(X) \cdot \pi(z, X)} \right] + \mathrm{Var}[\mu_z(X)] + 2\underbrace{\mathrm{Cov}\left[ \frac{Z \cdot R \cdot [Y - \mu_z(X)]}{e_z(X) \cdot \pi(z, X)}, \mu_z(X) \right]}_{=0} \\
=& \mathbb{E}\left[ \left[ \frac{Z \cdot R \cdot [Y - \mu_z(X)]}{e_z(X) \cdot \pi(z, X)} \right]^2 \right] - \left[ \frac{Z \cdot R \cdot}{e_z(X) \cdot \pi(z, X)} \underbrace{[\mathbb{E}[Y|Z, R = 1, X] - \mu_z(X)]}_{=0} \right]^2 \\
&+ \mathbb{E}[\mu_z(X)^2] - \mathbb{E}[\mu_z(X)]^2 \\
=& \mathbb{E}\left[ \left[ \frac{Z^2 \cdot R^2}{e_z^2(X) \cdot \pi^2(z, X)} \cdot [Y - \mu_z(X)]^2 \right] \right] + \mathbb{E}[\mu_z(X)^2] - \mathbb{E}[\mu_z(X)]^2 \\
=& \mathbb{E}\left[ \frac{Z \cdot R}{e_z^2(X) \cdot \pi^2(z, X)} \cdot [Y - \mu_z(X)]^2 \right] + \mathbb{E}[\mu(z, 1, X)^2] - \mathbb{E}[\mu_z(X)]^2 \\
=& \mathbb{E}\left[ \frac{1}{e_z(X) \cdot \pi(z, X)} \cdot [Y - \mu_z(X)]^2 \right] + \mathbb{E}[\mu_z(X)^2] - \mathbb{E}[\mu_z(X)]^2
\end{aligned}
$$

688 Lastly, $V_1 = V_2$. So the full variance term is

$$
\begin{aligned}
\mathrm{Var}[\psi_z - \psi_{z'}] &= \mathbb{E}\left[\frac{1}{e_z(X) \cdot \pi(z, X)} \cdot [Y - \mu_z(X)]^2\right] + \mathbb{E}\left[\frac{1}{e_{z'}(X) \cdot \pi(z', X)} \cdot [Y - \mu_{z'}(X)]^2\right] \\
&\quad + \mathbb{E}\left[(\mu_z(X) - \mu_{z'}(X))^2\right] - \mathbb{E}\left[\mu_z(X) - \mu_{z'}(X)\right]^2 \\
&= \mathbb{E}\left[\frac{1}{e_z(X) \cdot \pi(z, X)} \cdot [Y - \mu_z(X)]^2\right] + \mathbb{E}\left[\frac{1}{e_{z'}(X) \cdot \pi(z', X)} \cdot [Y - \mu_{z'}(X)]^2\right] \\
&\quad + \mathrm{Var}\left[\mu_z(X) - \mu_{z'}(X)\right]
\end{aligned}
$$

689 Rewriting the bound from Hahn (1998), we get

$$
\begin{aligned}
V \geq{} & \mathbb{E}\left[\frac{1}{e_z(X) \cdot \pi(z, X)} \cdot [Y - \mu_z(X)]^2\right] + \mathbb{E}\left[\frac{1}{e_{z'}(X) \cdot \pi(z', X)} \cdot [Y - \mu_{z'}(X)]^2\right] \\
& + \mathrm{Var}\left[\mu_z(X) - \mu_{z'}(X)\right]
\end{aligned}
$$

690 $\qquad\qquad\qquad\qquad\qquad\qquad\qquad\qquad\qquad\qquad\qquad\qquad\qquad\qquad\qquad\quad$ $\square$

691 *Proof of Theorem 4 .* Finding the optimal $\pi$ can be separated into sub-problems for each treatment
692 $z \in \{0, 1\}$, since the objective and dual variables are separable across $z$. We first look at a solution
693 for $\pi(z, X)$ for a given $z$:

$$
\begin{aligned}
\min_{\pi(z, x)} \ & \mathbb{E}\left[\frac{\sigma_z^2(X)}{e_z(X)\pi(z, X)}\right] & \text{(z-budget)} \\
\text{s.t. } & \mathbb{E}\left[\pi(z, X) \mid Z = z\right] \leq B_z, \\
& 0 < \pi(z, x) \leq 1, \ \forall x
\end{aligned}
$$

We define the Lagrangian of the optimization problem and introduce dual variables $\lambda$ for the budget constraint and $\eta$ and $\nu$ for the the constraint that $0 < \pi(z, X) \leq 1$:

$$
\mathcal{L} = \mathbb{E}\left[\frac{(Y - \mu_z(X))^2}{e_z(X)\pi(z, X)}\right] + \lambda_z(\mathbb{E}\left[\pi(z, X) \mid Z = z\right] - B_z) + \sum_{x \in \mathcal{X}}(\nu_x^z(\pi(z, x) - 1) - \eta_x^z\pi(z, x))
$$

Define the conditional outcome variance $\sigma^2(X) = \mathbb{E}\left[(Y - \mu(z, 1, X))^2 \mid X\right]$. Note that by iterated expectations,

$$
\mathcal{L} = \mathbb{E}\left[\frac{\sigma_z^2(X)}{e_z(X)\pi(z, X)}\right] + \lambda_z(\mathbb{E}\left[\pi(z, X) \mid Z = z\right] - B_z) + \sum_{x \in \mathcal{X}}(\nu_x^z(\pi(z, x) - 1) - \eta_x^z\pi(z, x))
$$

694 We can find the optimal solution by setting the derivative equal to 0. Since $p(X = x \mid Z = z) =$
695 $\frac{e_z(x)p(x)}{p(Z=z)}$

$$
\begin{aligned}
\frac{\partial \mathcal{L}}{\partial \pi(z, X)} &= -\frac{\sigma^2(X)}{e_z(X)(\pi^2(z, X))}p(x) + \lambda_z \frac{e_z(x)p(x)}{p(Z = z)} + \nu_x - \eta_x = 0, \text{ where } p(x) > 0 \\
&= -\frac{\sigma^2(X)}{e_z^2(X)\pi^2(z, X)} + \frac{\lambda_z}{p(Z = z)} + \frac{(\nu_x^z - \eta_x^z)}{p(x)e_z(x)} = 0
\end{aligned}
$$

Therefore

$$
\pi(z, x) = \sqrt{\frac{\sigma^2(x)}{e_z^2(x)\left(\frac{\lambda_z}{p(Z=z)} + \frac{(\nu_x^z - \eta_x^z)}{p(x)e_z(x)}\right)}}
$$

Next we give a choice of $\lambda$ that results in an interior solution with $0 \leq \pi(z, x) \leq 1$, so that $\nu_x^z, \eta_x^z$ can be set to $0$ without loss of generality to satisfy complementary slackness.

We posit a closed form solution

$$\pi^*(z, X) = \frac{\sqrt{\sigma_z^2(X)/e_z^2(X)}}{\mathbb{E}\left[\sqrt{\sigma_z^2(X)/e_z^2(X)} \mid Z = z\right]} \cdot B_z$$

.

Note that this solution is self-normalized to satisfy the budget constraint such that

$$\mathbb{E}\left[\pi^*(z, X)\mathbb{I}[Z = z]\right] = \mathbb{E}\left[\frac{\sqrt{\sigma^2(X)/e_z^2(X)}}{\mathbb{E}\left[\sqrt{\sigma_z^2(X)/e_z^2(X)} \mid Z = z\right]} B_z \mid Z = z\right] = B_z$$

This solution corresponds to a choice of $\lambda_z^* = {}^{p(Z=z)\mathbb{E}\left[\sqrt{\mathbb{I}[Z=z]\sigma^2(X)/e_z^2(X)}\right]^2}/B_z^2$ in the prior parametrized expression.

$$\pi_\lambda(z, X) = \pi^*(z, X)$$

$$\sqrt{\frac{\sigma_z^2(X)}{e_z^2(X)\frac{\lambda}{p(Z=z)}}} = \frac{\sqrt{\sigma_z^2(X)/e_z^2(X)}}{\mathbb{E}\left[\sqrt{\sigma_z^2(X)/e_z^2(X)} \mid Z = z\right]} \cdot B_z$$

We can check that the KKT conditions are satisfied at $\pi^*(z, X)$ and $\lambda^*$. We note that since $\pi^*(z, X)$ is an interior solution then w.l.o.g we can fix $\nu_x, \eta_x = 0$ to satisfy complementary slackness.

It remains to check that $\frac{\partial \mathcal{L}}{\partial \pi^*(z, X)} = 0$, we have that:

$$\frac{\partial \mathcal{L}}{\partial \pi(z, X)} = -\frac{\sigma_z^2(X)}{e_z(X)} \cdot \frac{e_z^2(X)\mathbb{E}\left[\sqrt{\sigma_z^2(X)/e_z(X)} \mid Z = z\right]^2}{\sigma_z^2(X) \cdot B_z^2} + \frac{\mathbb{E}\left[\sqrt{\sigma^2(X)/e_z(X)} \mid Z = z\right]^2 \sigma_z^2(X)e_z(X)}{\sigma_z^2(X) \cdot B_z^2} + 0 = 0.$$

Thus we have shown that $\pi^*(z, X)$ is optimal.

$\square$

*Proof of Theorem 1*. Proceed as in the proof of Theorem 4.

The Lagrangian of the optimization problem (with a single global budget constraint) is:

$$\mathcal{L} = \sum_{z \in \{0,1\}} \mathbb{E}\left[\frac{(Y - \mu_z(X))^2}{e_z(X)\pi(z, X)}\right] + \sum_{x \in \mathcal{X}}(\nu_x^z(\pi(z, x) - 1) - \eta_x^z\pi(z, x))$$

$$+ \lambda(\mathbb{E}\left[\pi(1, X)\mathbb{I}[Z = 1] + \pi(0, X)\mathbb{I}[Z = 0]\right] - B)$$

Again by iterated expectations,

$$\mathcal{L} = \mathbb{E}\left[\frac{\sigma_z^2(X)}{e_z(X)\pi(z, X)}\right] + \lambda(\mathbb{E}\left[\pi(1, X)e_1(X) + \pi(0, X)e_0(X)\right] - B_z) + \sum_{x \in \mathcal{X}}(\nu_x^z(\pi(z, x) - 1) - \eta_x^z\pi(z, x))$$

We can find the optimal solution by setting the derivative equal to $0$.

$$\frac{\partial \mathcal{L}}{\partial \pi(z, X)} = -\frac{\sigma^2(X)}{e_z(X)(\pi^2(z, X))}p(x) + \lambda p(x)e_z(x) + \nu_x^z - \eta_x^z = 0, \text{ where } p(x) > 0$$

$$= -\frac{\sigma^2(X)}{e_z^2(X)\pi^2(z, X)} + \lambda + \frac{(\nu_x^z - \eta_x^z)}{p(x)e_z(x)} = 0$$

Therefore we obtain a similar expression parametrized in $\lambda$, but this parameter is the same across both groups under a global budget.

$$\pi(z, x) = \sqrt{\frac{\sigma^2(x)}{e_z^2(x)(\lambda + \frac{(\nu_x^z - \eta_x^z)}{p(x)e_z(x)})}}$$

We can similarly give a closed-form expression for a different choice of $\lambda$ yielding an interior solution, so that we can set $\nu_x^z, \eta_x^z = 0$ without loss of generality.

$$\lambda = \frac{\mathbb{E}\left[\mathbb{I}[Z = 1]\sqrt{\sigma_1^2(X)/e_1^2(X)} + \mathbb{I}[Z = 0]\sqrt{\sigma_0^2(X)/e_0^2(X)}\right]^2}{B^2}$$

Notice that this satisfies the normalization requirement that $\mathbb{E}[\pi^\lambda(1, X)\mathbb{I}[Z = 1] + \pi^\lambda(0, X)\mathbb{I}[Z = 0]] \leq B$, and similarly note that the partial derivatives with respect to $\pi(z, x)$ are 0. $\qquad\square$

*Proof of Theorem 3* . The objective function arises from the asymptotic variance expression in [14, Thm. 3]; it follows readily from following their proof of Thm. 3 with our analysis of the asymptotic variance as in Proposition 1. The proof of the optimal solution follows our analysis in Theorem 1 with a few slightly different expressions, discussed as follows.

Then the Lagrangian is

$$\int \frac{\sigma^2(z \mid x)}{e(z, x)\pi(z, x)} f(x) dx + \lambda \left( \int \int \pi(z, x) K_h(z' - z) e(z', x) dz' f(x) dx \right)$$

Define the kernel localization of $e(z, x)$ around $z$ under the kernel function $K_h(z' - z)$:

$$\tilde{e}_h(z, x) = \int K_h(z' - z) e(z', x) dz'$$

Taking derivatives with respect to $\pi(z, x)$, we obtain the FOC

$$\nabla_{\pi(t\mid x)}\mathcal{L} = \frac{-\sigma^2(z, x)}{e(z, x)\pi(z, x)^2} f(x) + \lambda \tilde{e}_h(z, x) f(x) = 0$$

Solving the FOC, we obtain

$$\frac{-\sigma^2(z, x)}{e(z, x)\pi(z, x)^2} + \lambda \tilde{e}_h(z, x) = 0 \implies \pi^*(z, x) = \frac{1}{\lambda} \frac{\sqrt{\sigma^2(z, x)}}{e(z, x)} \sqrt{\frac{e(z, x)}{\tilde{e}_h(z, x)}}$$

We conclude that

$$\pi^*(z, x) \propto \frac{\sqrt{\sigma^2(z, x)}}{e(z, x)} \sqrt{\frac{e(z, x)}{\tilde{e}_h(z, x)}}$$

$\qquad\square$

*Proof of Theorem 2* . **Proof sketch.**

The proof proceeds in two steps. The first establishes that the feasible AIPW estimator converges to the AIPW estimator with oracle nuisances. It follows from standard analysis with cross-fitting, in particular the variant used across batches.

**Preliminaries** In the analysis, we write the score function as a function of $R$ in addition to other nuisance functions:

$$\psi_{z,i}(R_i, e, \pi, \mu) = \frac{\mathbb{I}[Z_i = z]R_i(Y_i - \mu_z(X_i))}{e_z(X_i)\pi(z, X_i)} + \mu_z(X_i)$$

The AIPW estimator can be rewritten as a sum over estimators within batch-$t$, fold-$k$, $\hat{\tau}_{AIPW}^{(t,k)}$, as follows:

$$\hat{\tau}_{AIPW} = \sum_{t=1}^{2}\sum_{k=1}^{K}\frac{n_{t,k}}{n}\sum_{(t,i)\in\mathcal{I}_k}\frac{1}{n_{t,k}}\{\hat{\psi}_{1,i}(R,\hat{e},\hat{\pi},\hat{\mu}) - \hat{\psi}_{0,i}(R,\hat{e},\hat{\pi},\hat{\mu})\} = \sum_{t=1}^{2}\sum_{k=1}^{K}\frac{n_{t,k}}{n}\hat{\tau}_{AIPW}^{(t,k)}$$

We introduce an intermediate quantity. The realized treatments are sampled with probability $\hat{\pi}(X_i)$, $R_i \sim Bern(\hat{\pi}(Z_i, X_i))$. In the asymptotic framework, we study treatments sampled from a mixture distribution over the two batches, $\tilde{R}_i \sim Bern(\pi^*(Z_i, X_i))$.

$$\tilde{\tau}_{AIPW} = \sum_{t=1}^{2}\sum_{k=1}^{K}\frac{n_{t,k}}{n}\sum_{(t,i)\in\mathcal{I}_k}\frac{1}{n_{t,k}}\{\hat{\psi}_{1,i}(\tilde{R},\hat{e},\hat{\pi},\hat{\mu}) - \hat{\psi}_{0,i}(R,\hat{e},\hat{\pi},\hat{\mu})\}$$

We also denote the AIPW estimator with oracle nuisances, $\hat{\tau}_{AIPW}^*$, as

$$\hat{\tau}_{AIPW}^* = \sum_{t=1}^{2}\sum_{k=1}^{K}\frac{n_{t,k}}{n}\sum_{(t,i)\in\mathcal{I}_k}\frac{1}{n_{t,k}}\{\psi_{1,i}(\tilde{R}_i,e,\pi,\mu) - \psi_{0,i}(\tilde{R}_i,e,\pi,\mu)\}$$

We study convergence within a batch$-t$, fold$-k$ subset; the decompositions above give that convergence also holds for the original estimators.

The first step studies the limiting mixture distribution propensity arising from the two-batch process and shows that the use of the double-machine learning estimator (AIPW), under the weaker product error assumptions, gives that the oracle estimator is asymptotically equivalent to the oracle estimator where missingness follows the limiting mixture missingness probability. The latter of these is a sample average of iid terms and follows a standard central limit theorem. Recalling that $\tilde{R}_i = \mathbb{I}[U_i \geq \pi^*(X_i)]$, we wish to show:

$$\sum_z \mathbb{E}_n[\psi_{z,i}(R,\hat{e},\hat{\pi},\hat{\mu})] - \mathbb{E}_n[\psi_{z,i}(\tilde{R},e,\pi,\mu)] = o_p(n^{-\frac{1}{2}}).$$

Next we show that the estimator with feasible nuisance estimators converges to the estimator with oracle knowledge of the nuisance functions

$$\sqrt{n}(\tilde{\tau}_{AIPW}^{(t,k)} - \hat{\tau}_{AIPW}^{*,(t,k)}) \to_p 0.$$

The result follows by the standard limit theorem applied to the estimator with oracle nuisance functions.

**Step 1**

Let $\tilde{R}_i = \mathbb{I}[U_i \geq \pi^*(Z_i, X_i)]$. Restricting attention to a single treatment value $z \in \{0,1\}$, we want to show that:

$$\sum_{t=1}^{2}\sum_{k=1}^{K}\frac{n_{t,k}}{n}\sum_{(t,i)\in\mathcal{I}_k}\frac{1}{n_{t,k}}\left\{\hat{\psi}_{1,i}(\tilde{R},\hat{e},\hat{\pi},\hat{\mu}) - \hat{\psi}_{1,i}(R,\hat{e},\hat{\pi},\hat{\mu})\right\}$$

$$= \sum_{t=1}^{2}\sum_{k=1}^{K}\frac{n_{t,k}}{n}\sum_{(t,i)\in\mathcal{I}_k}\frac{1}{n_{t,k}}\left\{\frac{\mathbb{I}[Z_i=z]\tilde{R}_i(Y_i - \hat{\mu}_z(X_i))}{\hat{e}_z(X_i)\hat{\pi}(z,X_i)} - \frac{\mathbb{I}[Z_i=z]R_i(Y_i - \hat{\mu}_z(X_i))}{\hat{e}_z(X_i)\hat{\pi}(z,X_i)}\right\} = o_p(n^{-1/2}).$$

Without loss of generality we further consider one summand on batch-$t$, fold-$k$ data, the same argument will apply to the other summands and the final estimator.

Note that by consistency of potential outcomes, for any data point we have that

$$\frac{\mathbb{I}[Z_i=z]\tilde{R}_i(Y_i - \hat{\mu}_z(X_i))}{\hat{e}_z(X_i)\hat{\pi}(z,X_i)} - \frac{\mathbb{I}[Z_i=z]R_i(Y_i - \hat{\mu}_z(X_i))}{\hat{e}_z(X_i)\hat{\pi}(z,X_i)} = \frac{\mathbb{I}[Z_i=z](\tilde{R}_i - R_i)(Y_i(z) - \hat{\mu}_z(X_i))}{\hat{e}_z(X_i)\hat{\pi}(z,X_i)}$$

For each batch $t = 1, \dots, T$ and fold $k = 1, \dots, K$, according to the CSBAE crossfitting procedure, we observe that conditional on $\mathcal{I}_{(-k)}$ for a given batch and the observed covariates, the summands

733 (namely $R_i = \mathbb{I}[U_i \leq \hat{\pi}^{(-k)}(X_i)]$) are independent mean-zero. The final estimator will consist of
734 the sum over batches and folds. We start by looking at the estimator over one batch $t$ and one fold $k$
735 and the rest follows for the other batches and folds.

$$
\frac{1}{n_{t,k}} \sum_{(t,i)\in\mathcal{I}_k} \frac{\mathbb{I}[Z_i = z](\tilde{R}_i - R_i)(Y_i(z) - \hat{\mu}_z(X_i))}{\hat{e}_z(X_i)\hat{\pi}(z, X_i)}
$$

$$
= \frac{1}{n_{t,k}} \sum_{(t,i)\in\mathcal{I}_k} \frac{\mathbb{I}[Z_i = z]\left((\tilde{R}_i - \pi^*(z, X_i)) + (\pi^*(z, X_i) - \hat{\pi}(z, X_i)) + (\hat{\pi}(z, X_i) - R_i)\right)(Y_i(z) - \hat{\mu}_z(X_i))}{\hat{e}_z(X_i)\hat{\pi}(z, X_i)}
$$

$$
\leq \nu_e\gamma_{\sigma^2} \frac{1}{n_{t,k}} \sum_{(t,i)\in\mathcal{I}_k} \mathbb{I}[Z_i = z]\left((\tilde{R}_i - \pi^*(z, X_i)) + (\pi^*(X_i) - \hat{\pi}(z, X_i)) + (\hat{\pi}(z, X_i) - R_i)\right)(Y_i(z) - \hat{\mu}_z(X_i))
$$

736 Applying Cauchy-Schwarz to each of these terms, we obtain product error rate terms. For the second
737 term, we obtain that

$$
\nu_e\gamma_{\sigma^2} \frac{1}{n_{t,k}} \sum_{(t,i)\in\mathcal{I}_k^z} (\pi^*(X_i) - \hat{\pi}(X_i))(Y_i(z) - \hat{\mu}_z(X_i))
$$

$$
\leq \nu_e\gamma_{\sigma^2} \sqrt{\frac{1}{n_{t,k}} \sum_{(t,i)\in\mathcal{I}_k^z} (\pi^*(X_i) - \hat{\pi}(X_i))^2} \sqrt{\frac{1}{n_{t,k}} \sum_{(t,i)\in\mathcal{I}_k^z} (Y_i(z) - \hat{\mu}_z(X_i))^2}
$$

$$
= \nu_e\gamma_{\sigma^2} \|\pi^*(X_i) - \hat{\pi}(X_i)\|_{2,n} \|Y_i(z) - \hat{\mu}_z(X_i)\|_{2,n}
$$

$$
= o_p(n^{-\frac{1}{2}}) \qquad\qquad\qquad\qquad (\text{Assumption 7})
$$

738 Analogously, we conclude that the first and third terms are $o_p(n^{-\frac{1}{2}})$, applying Cauchy-Schwarz to
739 each of them in turn.

740 **Step 2 (feasible estimator converges to oracle)**

741 If we look at one term for one treatment and datapoint in the above (the rest follows for the others),
742 we obtain the following decomposition into error and product-error terms:

$$
\frac{Z_i\tilde{R}_i(Y_i - \hat{\mu}_1(X_i))}{\hat{e}_1(X_i)\hat{\pi}(1, X_i)} - \frac{Z_i\tilde{R}_i(Y_i - \mu_1(X_i))}{e_1(X_i)\pi(1, X_i)} + (\hat{\mu}_1(X_i) - \mu_1(X_i))
$$

$$
= (\mu_1(X_i) - \hat{\mu}_1(X_i))\left(\frac{Z_i\tilde{R}_i}{e_1(X_i)\pi(1, X_i)} - 1\right) + Z_i\tilde{R}_i(Y_i - \hat{\mu}_1(X_i))(\frac{1}{\hat{e}_1(X_i)\hat{\pi}(1, X_i)} - \frac{1}{e_1(X_i)\pi(1, X_i)})
$$

$$
(\text{by } \pm \tfrac{Z_i\tilde{R}_i(Y_i - \hat{\mu}_1(X_i))}{e_1(X_i)\pi(1,X_i)})
$$

$$
= (\mu_1(X_i) - \hat{\mu}_1(X_i))\left(\frac{Z_i\tilde{R}_i}{e_1(X_i)\pi(1, X_i)} - 1\right) + Z_i\tilde{R}_i(Y_i - \mu_1(X_i))(\frac{1}{\hat{e}_1(X_i)\hat{\pi}(1, X_i)} - \frac{1}{e_1(X_i)\pi(1, X_i)})
$$

$$
+ Z_i\tilde{R}_i(\mu_1(X_i) - \hat{\mu}_1(X_i))(\frac{1}{\hat{e}_1(X_i)\hat{\pi}(1, X_i)} - \frac{1}{e_1(X_i)\pi(1, X_i)})
$$

$$
(\text{by } \pm Z_i\tilde{R}_i\mu_1(X_i)(\tfrac{1}{\hat{e}_1(X_i)\hat{\pi}(1,X_i)} - \tfrac{1}{e_1(X_i)\pi(1,X_i)}))
$$

$$
= (\mu_1(X_i) - \hat{\mu}_1(X_i))\left(\frac{Z_i\tilde{R}_i}{e_1(X_i)\pi(1, X_i)} - 1\right)
$$

$$
+ Z_i\tilde{R}_i(Y_i - \mu_1(X_i))\left(\hat{\pi}(1, X_i)^{-1}(\hat{e}_1(X_i)^{-1} - e_1(X_i)^{-1}) + e_1(X_i)^{-1}(\hat{\pi}(1, X_i)^{-1} - \pi(1, X_i)^{-1})\right)
$$

$$
+ Z_i\tilde{R}_i(\mu_1(X_i) - \hat{\mu}_1(X_i))\left(\hat{\pi}(1, X_i)^{-1}(\hat{e}_1(X_i)^{-1} - e_1(X_i)^{-1}) + e_1(X_i)^{-1}(\hat{\pi}(1, X_i)^{-1} - \pi(1, X_i)^{-1})\right)
$$

$$
(\text{by } \pm \tfrac{1}{\hat{e}\hat{\pi}})
$$

We want to show that

$$
\sqrt{n_{t,k}}(\hat{\tau}_{AIPW}^{(t,k)} - \hat{\tau}_{AIPW}^{*,(t,k)}) \to_p 0
$$

Now that we have written out this expansion for one datapoints, we can write out this expansion within a batch-$t$, fold-$k$ subset, and write out the cross-fitting terms for reference:

$$\sqrt{n_{t,k}}\left(\hat{\tau}_{AIPW}^{(t,k)} - \hat{\tau}_{AIPW}^{*,(t,k)}\right)$$

$$= \frac{1}{\sqrt{n_{t,k}}} \sum_{i:(t,i)\in\mathcal{I}_k} (\mu_1(X_i) - \hat{\mu}_1^{(-k)}(1,X_i))\left(\frac{Z_i\tilde{R}_i}{e_1(X_i)\pi(1,X_i)} - 1\right)$$

$$+ \frac{1}{\sqrt{n_{t,k}}} \sum_{i:(t,i)\in\mathcal{I}_k} Z_i\tilde{R}_i(Y_i - \mu_1(X_i))\times$$

$$\left(\hat{\pi}^{(-k)}(1,X_i)^{-1}(\hat{e}_1^{(-k)}(X_i)^{-1} - e_1(X_i)^{-1}) + e_1(X_i)^{-1}(\hat{\pi}^{(-k)}(1,X_i)^{-1} - \pi(1,X_i)^{-1})\right)$$

$$+ \frac{1}{\sqrt{n_{t,k}}} \sum_{i:(t,i)\in\mathcal{I}_k} Z_i\tilde{R}_i(\mu_1(X_i) - \hat{\mu}_1^{(-k)}(1,X_i))\times$$

$$\left(\hat{\pi}^{(-k)}(1,X_i)^{-1}(\hat{e}_1^{(-k)}(X_i)^{-1} - e_1(X_i)^{-1}) + e_1(X_i)^{-1}(\hat{\pi}^{(-k)}(1,X_i)^{-1} - \pi(1,X_i)^{-1})\right)$$

**Bound for third term**:

$$\frac{1}{\sqrt{n_{t,k}}} \sum_{i:(t,i)\in\mathcal{I}_k} Z_i\tilde{R}_i(\mu_1(X_i) - \hat{\mu}_1^{(-k)}(X_i))(\hat{\pi}^{(-k)}(1,X_i)^{-1}(\hat{e}_1^{(-k)}(X_i)^{-1} - e_1(X_i)^{-1})$$

$$+ e_1(X_i)^{-1}(\hat{\pi}^{(-k)}(1,X_i)^{-1} - \pi(1,X_i)^{-1})$$

$$= \frac{1}{\sqrt{n_{t,k}}} \sum_{i:(t,i)\in\mathcal{I}_k} Z_i\tilde{R}_i\hat{\pi}^{(-k)}(1,X_i)^{-1}(\mu_1(X_i) - \hat{\mu}_1^{(-k)}(X_i))(\hat{e}_1^{(-k)}(X_i)^{-1} - e_1(X_i)^{-1})$$

$$+ Z_i\tilde{R}_i e_1(X_i)^{-1}(\mu_1(X_i) - \hat{\mu}_1^{(-k)}(X_i))(\hat{\pi}^{(-k)}(1,X_i)^{-1} - \pi(1,X_i)^{-1})$$

$$\leq (\lambda_\pi + \nu_e)\frac{1}{\sqrt{n_{t,k}}} \sum_{i:(t,i)\in\mathcal{I}_k} (\mu_1(X_i) - \hat{\mu}_1^{(-k)}(X_i))(\hat{e}_1^{(-k)}(X_i)^{-1} - e_1(X_i)^{-1})$$

$$+ (\mu_1(X_i) - \hat{\mu}_1^{(-k)}(X_i))(\hat{\pi}^{(-k)}(1,X_i)^{-1} - \pi(1,X_i)^{-1})$$

$$\leq (\lambda_\pi + \nu_e)\delta_n n^{-1/2}$$

where the last inequality makes use of product error rate assumptions 5-6 and nuisance function convergence rates from Lemma 4. Thus, we find that this term is $o_p(1/\sqrt{n})$

**Bound for the first term**:

The key to bounding the first term is that cross-fitting allows us to treat this term as the average of independent mean-zero random variables. We will bound it with Chebyshev's inequality, which requires a bound on the second moment on the summands in the first term.

$$\mathbb{E}\left[\frac{1}{\sqrt{n_{t,k}}}\sum_{i:(t,i)\in\mathcal{I}_k}\left((\mu_1(X_i)-\hat{\mu}_1^{(-k)}(X_i))\left(\frac{Z_i\tilde{R}_i}{e_1(X_i)\pi(1,X_i)}-1\right)\right)^2\mid\mathcal{I}_{(-k)},\{X_i\}\right]$$

$$=\mathrm{Var}\left[\frac{1}{\sqrt{n_{t,k}}}\sum_{i:(t,i)\in\mathcal{I}_k}(\mu_1(X_i)-\hat{\mu}_1^{(-k)}(X_i))\left(\frac{Z_i\tilde{R}_i}{e_1(X_i)\pi(1,X_i)}-1\right)\mid\mathcal{I}_{(-k)},\{X_i\}\right]$$

$$=\frac{1}{n_{t,k}}\sum_{i:(t,i)\in\mathcal{I}_k}\mathbb{E}\left[(\mu_1(X_i)-\hat{\mu}_1^{(-k)}(X_i))^2\left(\frac{Z_i\tilde{R}_i}{e_1(X_i)\pi(1,X_i)}-1\right)^2\mid\mathcal{I}_{(-k)},\{X_i\}\right]$$

$$\text{(expectation of }(\tfrac{Z_i\tilde{R}_i}{e_1(X_i)\pi(1,X_i)}-1)^2)$$

$$=\frac{1}{n_{t,k}}\sum_{i:(t,i)\in\mathcal{I}_k}\frac{1-e_1(X_i)\pi(z,X_i)}{e_1(X_i)\pi(1,X_i)}(\mu_1(X_i)-\hat{\mu}_1^{(-k)}(X_i))^2$$

$$\leq\frac{1-\nu_e\lambda_\pi}{\nu_e\lambda_\pi}\frac{1}{n_{t,k}}\sum_{i:(t,i)\in\mathcal{I}_k}((\mu_1(X_i)-\hat{\mu}_1^{(-k)}(X_i))^2=o_p(\frac{1}{n^{1+2r_\mu}})$$

752  where for the third equality, we use the fact that

753  $\mathbb{E}[(\frac{Z_i\tilde{R}_i}{e_1(X_i)\pi(1,X_i)}-1)^2\mid\mathcal{I}_{(-k)},\{X_i\}]=\mathbb{E}[(\frac{Z_i^2R_i^2}{e_1^2(X_i)\pi^2(1,X_i)}-2\frac{Z_i\tilde{R}_i}{e_1(X_i)\pi(1,X_i)}+1\mid\mathcal{I}_{(-k)},\{X_i\}]=\frac{1}{e_1(X_i)\pi(1,X_i)}-1$

754  Since $r_\mu\geq0$, we can conclude by Chebyshev's inequality that the first term is $o_p(n^{-1/2})$.

755  **Bound for the second term**: We bound the second term following a similar argument as above.

$$\mathbb{E}\left[\frac{1}{\sqrt{n_{t,k}}}\sum_{i:(t,i)\in\mathcal{I}_k}\left(Z_i\tilde{R}_i(Y_i-\mu_1(X_i))\left(\hat{\pi}^{(-k)}(1,X_i)^{-1}(\hat{e}_1^{(-k)}(X_i)^{-1}-e_1(X_i)^{-1})\right)^2\mid\mathcal{I}_{(-k)},\{X_i\}\right]\right.$$

$$+\mathbb{E}\left[\frac{1}{\sqrt{n_{t,k}}}\sum_{i:(t,i)\in\mathcal{I}_k}\left(Z_i\tilde{R}_i(Y_i-\mu_1(X_i))\left(e_1(X_i)^{-1}(\hat{\pi}^{(-k)}(1,X_i)^{-1}-\pi(1,X_i)^{-1})\right)\right)^2\mid\mathcal{I}_{(-k)},\{X_i\}\right]$$

$$=\mathrm{Var}\left[\frac{1}{\sqrt{n_{t,k}}}\sum_{i:(t,i)\in\mathcal{I}_k}\left(Z_i\tilde{R}_i(Y_i-\mu_1(X_i))\left(\hat{\pi}^{(-k)}(1,X_i)^{-1}(\hat{e}_1^{(-k)}(X_i)^{-1}-e_1(X_i)^{-1})\right)\mid\mathcal{I}_{(-k)},\{X_i\}\right]\right.$$

$$+\mathrm{Var}\left[\frac{1}{\sqrt{n_{t,k}}}\sum_{i:(t,i)\in\mathcal{I}_k}\left(Z_i\tilde{R}_i(Y_i-\mu_1(X_i))\left(e_1(X_i)^{-1}(\hat{\pi}^{(-k)}(1,X_i)^{-1}-\pi(1,X_i)^{-1})\right)\mid\mathcal{I}_{(-k)},\{X_i\}\right]\right.$$

$$=\frac{1}{n_{t,k}}\sum_{i:(t,i)\in\mathcal{I}_k}\mathbb{E}\left[\left(\hat{\pi}^{(-k)}(1,X_i)^{-1}(\hat{e}_1^{(-k)}(X_i)^{-1}-e_1(X_i)^{-1})\right)^2\frac{Z_i^2R_i^2}{(\hat{\pi}^{(-k)}(1,X_i))^2}(Y_i-\mu_1(X_i))^2\mid\mathcal{I}_{(-k)},\{X_i\}\right]$$

$$+\frac{1}{n_{t,k}}\sum_{i:(t,i)\in\mathcal{I}_k}\mathbb{E}\left[\left(e_1(X_i)^{-1}(\hat{\pi}^{(-k)}(1,X_i)^{-1}-\pi(1,X_i)^{-1})\right)^2\frac{Z_i^2R_i^2}{(\hat{\pi}^{(-k)}(1,X_i))^2}(Y_i-\mu_1(X_i))^2\mid\mathcal{I}_{(-k)},\{X_i\}\right]$$

$$=\frac{1}{n_{t,k}}\sum_{i:(t,i)\in\mathcal{I}_k}\frac{e_1^2(X_i)\pi^2(z,X_i)}{(\hat{\pi}^{(-k)}(1,X_i))^2}\mathbb{E}[\sigma^2(X_i)\mid\mathcal{I}_{(-k)},\{X_i\}]\hat{e}_1^{(-k)}(X_i)^{-1}-e_1(X_i)^{-1}))^2$$

$$+\frac{e_1^2(X_i)(\pi^{(-k)}(z,X_i))^2}{e_1(X_i)}\mathbb{E}[\sigma^2(X_i)\mid\mathcal{I}_{(-k)},\{X_i\}](\hat{\pi}^{(-k)}(1,X_i)^{-1}-\pi(1,X_i)^{-1})^2$$

756

$$\leq\frac{1}{n_{t,k}}\sum_{i:(t,i)\in\mathcal{I}_k}\frac{\nu_e^2\lambda_\pi^2}{(\hat{\pi}^{(-k)}(1,X_i))^2}B_{\sigma^2}(\hat{e}_1^{(-k)}(X_i)^{-1}-e_1(X_i)^{-1}))^2+\frac{\nu_e^2\lambda_\pi^2}{\nu_e^2}B_{\sigma^2}(\hat{\pi}^{(-k)}(1,X_i)^{-1}-\pi(1,X_i)^{-1})^2$$

$$=o_p(\frac{1}{n^{1+2r_e+2r_\pi}})$$

757  where the last inequality is because $\sigma^2(X)$ is bounded above, $\sigma^2(X)\leq B_{\sigma^2}$, by Lemma 4. Thus, by
758  similar argument to the first term, since this term is a sum of zero-mean random variables and since

$r_\pi, r_e \geq 0$, we can apply Chebyshev's inequality and get that this term is also $o_p(1/\sqrt{n})$. This holds for both treatments. Therefore,

$$\sqrt{n_{t,k}}(\hat{\tau}_{AIPW}^{(t,k)} - \hat{\tau}_{AIPW}^{*,(t,k)}) \to_p 0.$$

Putting these results from Step 1 and Step 2 together, along with the fact that $\frac{n_{t,k}}{n} \to \frac{1}{K}$, gives the theorem. $\qquad\square$

# H  Additional Lemmas

## H.1  Results appearing in other works, stated for completeness.

**Lemma 1** (Conditional convergence implies unconditional convergence, from [10]). *Lemma 6.1. (Conditional Convergence implies unconditional) Let $\{X_m\}$ and $\{Y_m\}$ be sequences of random vectors. (a) If, for $\epsilon_m \to 0, \Pr(\|X_m\| > \epsilon_m \mid Y_m) \to_{\Pr} 0$, then $\Pr(\|X_m\| > \epsilon_m) \to 0$. In particular, this occurs if $E[\|X_m\|^q / \epsilon_m^q \mid Y_m] \to_{Pr} 0$ for some $q \geq 1$, by Markov's inequality. (b) Let $\{A_m\}$ be a sequence of positive constants. If $\|X_m\| = O_P(A_m)$ conditional on $Y_m$, namely, that for any $\ell_m \to \infty, \Pr(\|X_m\| > \ell_m A_m \mid Y_m) \to_{Pr} 0$, then $\|X_m\| = O_P(A_m)$ unconditionally, namely, that for any $\ell_m \to \infty, \Pr(\|X_m\| > \ell_m A_m) \to 0$.*

**Lemma 2** (Chebyshev's inequality). *Let $X$ be a random variable with mean $\mu$ and variance $\sigma^2$. Then, for any $t > 0$, we have*

$$P(|X - \mu| \geq t) \leq \frac{\sigma^2}{t^2}$$

**Lemma 3** (Theorem 8.3.23 (Empirical processes via VC dimension), [48]). *Let $\mathcal{F}$ be a class of Boolean functions on a probability space $(\Omega, \Sigma, \mu)$ with finite $VC$ dimension $\mathrm{vc}(\mathcal{F}) \geq 1$. Let $X, X_1, X_2, \ldots, X_n$ be independent random points in $\Omega$ distributed according to the law $\mu$. Then*

$$\mathbb{E} \sup_{f \in \mathcal{F}} \left| \frac{1}{n} \sum_{i=1}^{n} f(X_i) - \mathbb{E}f(X) \right| \leq C \sqrt{\frac{\mathrm{vc}(\mathcal{F})}{n}}$$

## H.2  Lemmas

**Lemma 4** (Convergence of $\hat{\pi}$). *Assume that with high probability, for some large constant $K$, $\|\hat{e}(X) - e(X)\|_2 \leq K n^{-r_e}, \|\hat{\sigma}^2(X) - \sigma^2(X)\|_2 \leq K n^{-r_\sigma}$. Assume Assumption 8. Assume that $\sigma^2(X) > 0$ so that its inverse is bounded $1/\sigma^2(X) \leq \gamma_\sigma$. Recall that Theorem 1 gives that*

$$\pi^*(z, X) = \sqrt{\frac{\sigma_z^2(X)}{e_z^2(X)}} B \left( \mathbb{E}\left[ \mathbb{I}[Z=1]\sqrt{\frac{\sigma_1^2(X)}{e_1^2(X)}} + \mathbb{I}[Z=0]\sqrt{\frac{\sigma_0^2(X)}{e_0^2(X)}} \right] \right)^{-1}$$

*Define $\hat{\pi}^*(z, x)$ to be a plug-in version of the above (with $\hat{\sigma}^2, \hat{e}$, and $\mathbb{E}_n[\cdot]$). Then*

$$\|\hat{\pi}^*(z, X) - \pi^*(z, X)\|_2 = o_p(n^{-\min(r_e, r_\sigma, 1/2)}).$$

*Proof.* Let $a = \frac{\sigma_z^2(X)}{e_z^2(X)}, b = \mathbb{E}\left[ \mathbb{I}[Z=1]\sqrt{\frac{\sigma_1^2(X)}{e_1^2(X)}} + \mathbb{I}[Z=0]\sqrt{\frac{\sigma_0^2(X)}{e_0^2(X)}} \right]$.

Let $c = \frac{\hat{\sigma}_z^2(X)}{\hat{e}_z^2(X)}, d = \mathbb{E}_n\left[ \mathbb{I}[Z=1]\sqrt{\frac{\hat{\sigma}_1^2(X)}{\hat{e}_1^2(X)}} + \mathbb{I}[Z=0]\sqrt{\frac{\hat{\sigma}_0^2(X)}{\hat{e}_0^2(X)}} \right]$.

Then $\|\pi^*(z, X) - \hat{\pi}^*(z, X)\|_2 = \|a/b - c/d\|_2$.

Positivity of $\sigma_z^2(X)$ gives the elementary equality that $\frac{a}{b} - \frac{c}{d} = \left(\frac{a-b}{b}\right) + \left(\frac{d-c}{d}\right)$.

Therefore, by triangle inequality and boundedness,

$$\|\pi^*(z,X) - \hat{\pi}^*(z,X)\|_2 \le \gamma_\sigma \left\| \sqrt{\sigma^2(X)/e^2(X)} - \sqrt{\hat{\sigma}^2(X)/\hat{e}^2(X)} \right\|_2$$

$$+ \gamma_\sigma \left| \mathbb{E}_n \left[ \mathbb{I}[Z=1]\sqrt{\frac{\hat{\sigma}_1^2(X)}{\hat{e}_1^2(X)}} + \mathbb{I}[Z=0]\sqrt{\frac{\hat{\sigma}_0^2(X)}{\hat{e}_0^2(X)}} \right] - \mathbb{E}\left[ \mathbb{I}[Z=1]\sqrt{\frac{\sigma_1^2(X)}{e_1^2(X)}} + \mathbb{I}[Z=0]\sqrt{\frac{\sigma_0^2(X)}{e_0^2(X)}} \right] \right|$$

$$(2)$$

Next we show that for $z \in \{0,1\}$,

$$\left\| \sqrt{\hat{\sigma}_z^2(X)/\hat{e}_z^2(X)} - \sqrt{\sigma_z^2(X)/e_z^2(X)} \right\|_2 \le \nu_e B_{\sigma^2}\left( \left\| \sqrt{\hat{\sigma}_z^2(X)} - \sqrt{\sigma_z^2(X)} \right\|_2 + \|e_z(X) - \hat{e}_z(X)\|_2 \right)$$

$$(3)$$

In the below, we drop the $z$ argument.

By the triangle inequality, boundedness of $1/\hat{e}(X) \le \nu_e$, and of $\sigma^2(X) \le B_{\sigma^2}$:

$$\left\| \sqrt{\hat{\sigma}^2(X)/\hat{e}^2(X)} - \sqrt{\sigma^2(X)/e^2(X)} \right\|_2$$

$$= \left\| \sqrt{\hat{\sigma}^2(X)/\hat{e}^2(X)} \pm \sqrt{\sigma^2(X)/\hat{e}^2(X)} - \sqrt{\sigma^2(X)/e^2(X)} \right\|_2$$

$$\le \nu_e \left\| \sqrt{\hat{\sigma}^2(X)} - \sqrt{\sigma^2(X)} \right\|_2 + B_{\sigma^2} \left\| \frac{1}{e(X)} - \frac{1}{\hat{e}(X)} \right\|_2$$

For the second term:

$$B_{\sigma^2} \left\| \frac{1}{e(X)} - \frac{1}{\hat{e}(X)} \right\|_2 \le B_{\sigma^2} \left\| \frac{1}{e(X)} - \frac{1}{\hat{e}(X)} \right\|_2 \le B_{\sigma^2}\nu_e \|e(X) - \hat{e}(X)\|_2$$

since $1/e(X)$ is Lipschitz on the assumed bounded domain (overlap assumption).

For the first term:

$$\nu \left\| \sqrt{\hat{\sigma}^2(X)} - \sqrt{\sigma^2(X)} \right\|_2 \le \nu_e B_{\sigma^2} \left\| \hat{\sigma}^2(X) - \sigma^2(X) \right\|_2$$

since $\sigma^2(X)$ is bounded away from 0, then $\sqrt{\sigma^2(X)}$ is Lipschitz.

This proves Equation (3), which bounds the first term of Equation (2). For the second term, denote for brevity

$$\hat{\beta}(\sigma, e) = \mathbb{E}_n \left[ \mathbb{I}[Z=1]\sqrt{\frac{\sigma_1^2(X)}{e_1^2(X)}} + \mathbb{I}[Z=0]\sqrt{\frac{\sigma_0^2(X)}{e_0^2(X)}} \right],$$

and $\beta(\sigma, e)$ to be the above with $\mathbb{E}[\cdot]$ instead of $\mathbb{E}_n[\cdot]$. Then the second term of Equation (2) is $\hat{\beta}(\hat{\sigma}, \hat{e}) - \beta(\sigma, e)$, and decomposing further, that

$$\hat{\beta}(\hat{\sigma}, \hat{e}) - \beta(\sigma, e) = \hat{\beta}(\hat{\sigma}, \hat{e}) - \hat{\beta}(\sigma, e) + \hat{\beta}(\sigma, e) - \beta(\sigma, e).$$

Note that by Cauchy-Schwarz inequality, and Lemma 3 (chaining with VC-dimension),

$$\hat{\beta}(\hat{\sigma}, \hat{e}) - \hat{\beta}(\sigma, e) \le 2\nu_e B_{\sigma^2} \left( \left\| \sqrt{\hat{\sigma}_z^2(X)} - \sqrt{\sigma_z^2(X)} \right\|_2 + \|e_z(X) - \hat{e}_z(X)\|_2 \right) + 2C\sqrt{\frac{\text{vc}(\mathcal{F}_{\sqrt{\frac{\sigma^2}{e}}})}{n}}$$

And another application of Lemma 3 gives that

$$\hat{\beta}(\sigma, e) - \beta(\sigma, e) = (\mathbb{E}_n - \mathbb{E})\left[ \mathbb{I}[Z=1]\sqrt{\frac{\sigma_1^2(X)}{e_1^2(X)}} + \mathbb{I}[Z=0]\sqrt{\frac{\sigma_0^2(X)}{e_0^2(X)}} \right] \le 2C\sqrt{\frac{\text{vc}(\mathcal{F}_{\sqrt{\frac{\sigma^2}{e}}})}{n}}.$$

Combining the above bounds with Equation (2), we conclude that $\|\pi^*(z,X) - \hat{\pi}^*(z,X)\|_2 = o_p(n^{-\min(r_e, r_\sigma, 1/2)})$. $\square$

# I Additional Experiment, Details and Discussion

## I.1 Additional details

All experiments using our full algorithm 2 were conducted on a 2021 13-inch MacBook Pro equipped with a 2.3 GHz Quad-Core Intel Core i7 processor and 32 GB of memory. This setup was used to train standard nuisance models using machine learning, evaluated our algorithm, and conduct the analysis tasks reported in this paper. The average compute time for the experiments on real world data with 20 trials was less than 30 minutes, while the simulated data with 100 trials took less than 60 minutes. Additionally, for all experiments, we allocate $55\%$ of the data to batch 1 and $45\%$ to batch 2.

We run the ML nuisance models, logistic regression, random forest and support vectors machines, using popular Python packages (i.e. sklearn and scipy). We use logistic regression to estimate the propensity scores. For the outcome and variance models, we use random forest with the following hyperparameters:

- max_depth: None
- min_samples_leaf: 4
- min_samples_split: 10
- n_estimators: 100
- random_state: 42

We also use SVM model for the outcome models incorporating LLM predictions, and we use the following hyperparameters:

- kernel: 'rbf'
- C: 1

We chose these hyperparameters by doing a grid search over hyperparameters and chose the ones that performed the best.

We run LLM calls on Together.AI since they provide enterprise-secure deployments of local models, which is required for sensitive data. Because we need to use local LLMs for the real-world street outreach data, we also use the same local LLMs for the other experiments. We use "Llama-3.3-70B-Instruct-Turbo" for all experiments using LLMs. (Larger models provide effectively similar performance).

To solve our optimization problem, we used the python package CVXPY and we specifically used the Splitting Conic Solver (SCS) solver.

Once the experiments are run, we display the means and $95\%$ confidence interval bands, obtained through bootstrapping, in each of our figures.

## I.2 Synthetic Data

Before running our batch adaptive algorithm, we split the data into a validation set ($35\%$ of data) in which we estimate the ATE on. Then we use the remainder of the data to run our algorithm, which splits that data into the two batches in the way we described previously.

**Data Generating Process.** We generate a dataset $\mathcal{D} = \{X, Z, Y, Y(1), Y(0)\}$, of size 1000 and where the true ATE $\tau = \mathbb{E}[Y(1)] - \mathbb{E}[Y(0)] = 3$. We sample each covariate $X \in \mathbb{R}^5$ from a standard normal distribution, $X \sim \mathcal{N}(0, I_5)$. Treatment $Z$ is drawn with logistic probability $\gamma_z(X) = (1 + e^{X_2 + X_3 + 0.5})$. We define $\sigma_z^2(X)$ as follows:

$$\sigma_1^2(X) := \max[1.3 + 0.4\sin(X_1), 0]$$
$$\sigma_0^2(X) := \max[3.5 + 0.3\cos(X_3), 0].$$

Finally, the outcome models are defined as:

$$Y(0) = 5 + X_1 - 2X_2 + \epsilon_0$$
$$Y(1) = Y(0) + \theta_0 + \epsilon_1,$$

where $\epsilon_0 \sim \mathcal{N}(0, \sigma_0(X))$ and $\epsilon_1 \sim \mathcal{N}(0, \sigma_1(X))$. The observed outcomes are $Y = Z \cdot Y(1) + (1 - Z) \cdot Y(0)$.

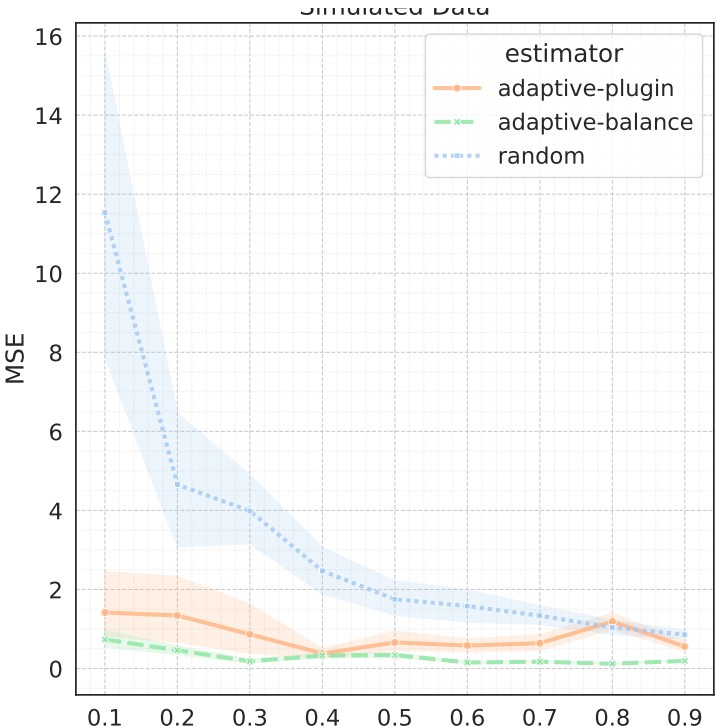

Figure 4: Mean squared error between estimated ATE and true ATE averaged over 100 trials across varying budgets.

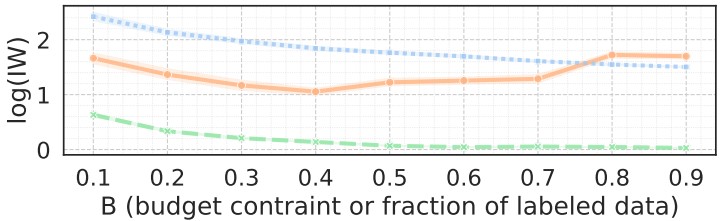

Figure 5: Average confidence interval width averaged over 100 trials across varying budgets.

**Results.** We see the greatest advantage with our adaptive estimation for budgets between 0.1 and 0.4. While for larger budgets, even as the MSE for both estimators converge, the interval width for the adaptive estimator is still relatively small. Adaptive annotation with a larger budget introduces additional variation in inverse annotation probabilities, as compared to uniform sampling, which is equivalent to full-information estimation at a marginally smaller budget. This regime of improvement for small budgets is nonetheless practically relevant and consistent with other works.

To stabilize the estimation of the inverse annotation probabilities, we use the plug-in estimator following eq. ($RZ$-plug-in.) and the ForestReisz method to estimate the balancing weights [11].This approach provides an automatic machine learning debiasing procedure to learn the Reisz representer, or unique weights that automatically balances functions between treated and control groups using a random forest model.

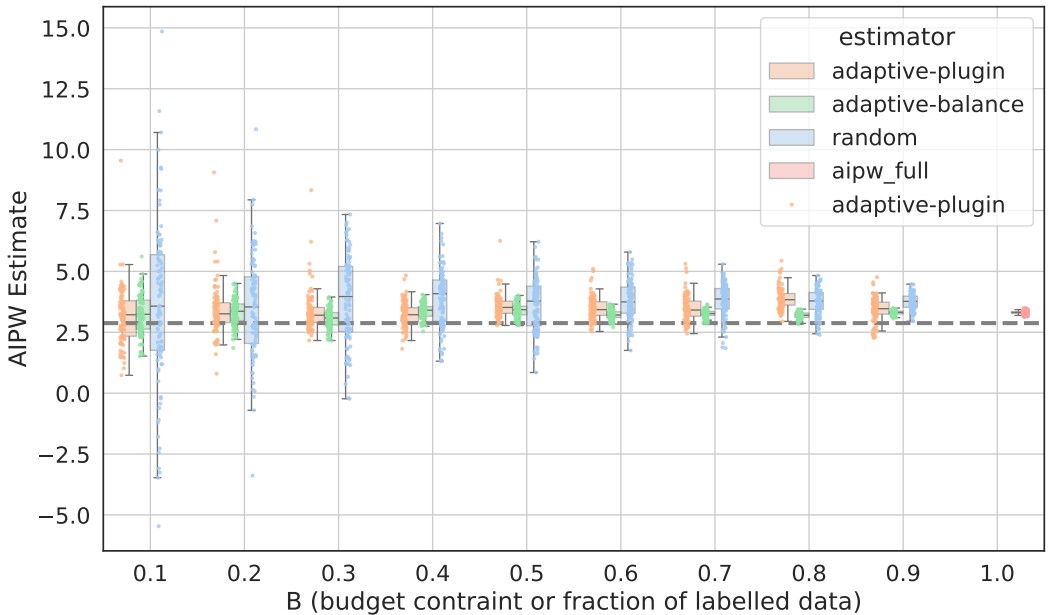

Figure 6: Boxplots of ATE estimates compared to skyline $\hat{\tau}_{AIPW}$ when the labeling budget is the entire dataset in red and the grey dotted line is $\tau$.

## I.3 Real-world Dataset Details

We provide further details about the treatment, covariates and outcomes for each dataset. Table 1 and table 2 describe the variables in the retail hero and outreach datasets, respectively. We refer the reader to [17] for further details about the dataset. For the outreach data, we constructed the binary treatment variable by binning the frequency of outreach engagements for each client within the first 6 months of the treatment period. We checked for overlap in propensity scores and decided to use treatments in the middle of the distribution as they had the most overlap. Additionally, by corollary 1, our method does well even when the propensity scores do not have good overlap.

| Variable | Description | Discrete Category |
|---|---|---|
| **Outcome** | | |
| Purchase | whether a customer purchased a product | [Yes,No] |
| **Treatment** | | |
| SMS communication | whether a text was sent to encourage customer to continue shopping | [Yes, No] |
| **Covariates** | | |
| avg. purchase | avg. purchase value per transaction | [1-263, 264-396, 397-611, > 612] |
| avg. product quantity | avg. number of products bought | [$\leq 7, > 7$] |
| avg. points received | avg. number of points received | [$\leq 5, > 5$] |
| num transactions | total number of transactions so far | [$\leq 8$, 9 - 15, 16 - 27, $> 28$] |
| age | age of user | [$\leq 45, > 45$] |

Table 1: Covariate, treatment, and outcome descriptions and discrete category definitions for Retail-Hero dataset.

## I.4 Additional Context on Street Outreach

In New York City alone, approximately $\$80,000,000$ per year is invested in homeless street outreach to an unclear effect. It is a time-consuming process, and it is unclear how the impacts of such intensive

| Variable | Description | Discrete Category |
|---|---|---|
| **Outcome** | | |
| Placement | The greatest housing placement attained by the client between 2019–2021 | [3:permanent housing, 2: shelter/transitional housing, 1: other (e.g., hospital), 0: streets] |
| **Treatment** | | |
| Street outreach | Binned frequency of outreach within the first three months of 2019 | [More outreach (3–15), Less outreach (1–2)] |
| **Covariates** | | |
| DateFirstSeen | Ordinal date when the client was first seen by the outreach team | NA |
| Program | Outreach or service program the client belonged to | [Brooklyn Library, Grand Central Partnership, Hospital to Home, K-Mart Alley, Macy's, MetLife, Penn Post Office, Pyramid Park, S2H Bronx, S2H Brooklyn, S2H Manhattan, S2H Queens, Starbucks, Superblock, Vornado, Williamsburg Stabilization Bed] |
| BelievedChronic | Perceived by outreach workers as chronically homeless individual | [Yes, No] |
| Gender | Perceived or disclosed gender of client | [Female, Male, Transgender] |
| Race | Perceived or disclosed race of client | [American Indian/Alaskan Native, Asian, Black/African American, Native Hawaiian/Pacific Islander, White/Caucasian] |
| Ethnicity | Perceived or disclosed ethnicity of client | [Hispanic/Latino, Non-hispanic/latino] |
| Age | Perceived or disclosed age range of client | [< 30 years old, 30–50 years old, > 50 years old] |
| Was311Call | Whether outreach workers were responding to a 311 city call | [Yes, No] |
| Was911Call | Whether 911 was called to the scene | [Yes, No] |
| Removal958 | Whether outreach workers were responding to removal hotline call | [Yes, No] |
| Housing application | Whether any mention of the housing application was found in casenotes | [Yes, No] |
| Service refusal | Whether outreach worker documented that a client refused their services in casenotes | [Yes, No] |
| Important documents | Whether there was mention of any important documents (i.e. social security card, drivers license, etc,) in casenotes | [Yes, No] |
| Benefits | Whether there was any mention of social service benefits in the casenotes (i.e. foodstamps, SSI) | [Yes, No] |
| num contacts | number of engagements with an outreach worker prior to 2019 | NA |
| max Placement | maximum housing placement reached before 2019 | [3:permanent housing, 2: shelter/transitional housing, 1: other (e.g., hospital), 0: streets] |

Table 2: Covariates, treatment, and outcome descriptions and discrete category definitions for the Street Outreach dataset.

individualized outreach might compare to other proposed approaches, such as those focusing on placing entire networks of individuals together. While the nonprofit reports key metrics such as number of completed placements in housing services, these can be somewhat rare due to length

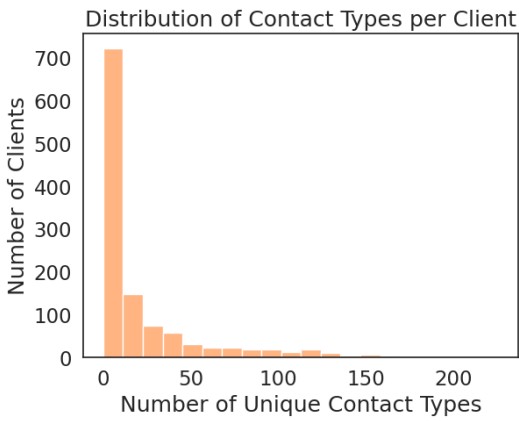

Figure 7: Distribution of street outreach engagements for client population.

of outreach, delays in waiting for housing, matching issues, etc; moreover, much of a successful placement is out of the control of outreach due to highly limited housing capacities. Measuring the impacts of street outreach on intermediate outcomes such as accessing benefits and services, completing required appointments and interviews, can better reflect the immediate impacts of street outreach.

## I.5    Robustness Check on Street Outreach Data

To further demonstrate the utility of our approach, we run experiments on the Street Outreach data with $\tilde{Y}$. To recap, our setup consists of covariates $X$, which includes client characteristics at baseline and LLM-generated summaries of case notes recorded before the treatment period. In the main text, we used LLMs to summarize casenotes prior to outreach during the interventional period, and used them in zero-shot prediction of later placement outcomes. Here we also incorporate LLM-generated summaries of case notes recorded post-treatment. These represent $\tilde{Y}$ in our framework.

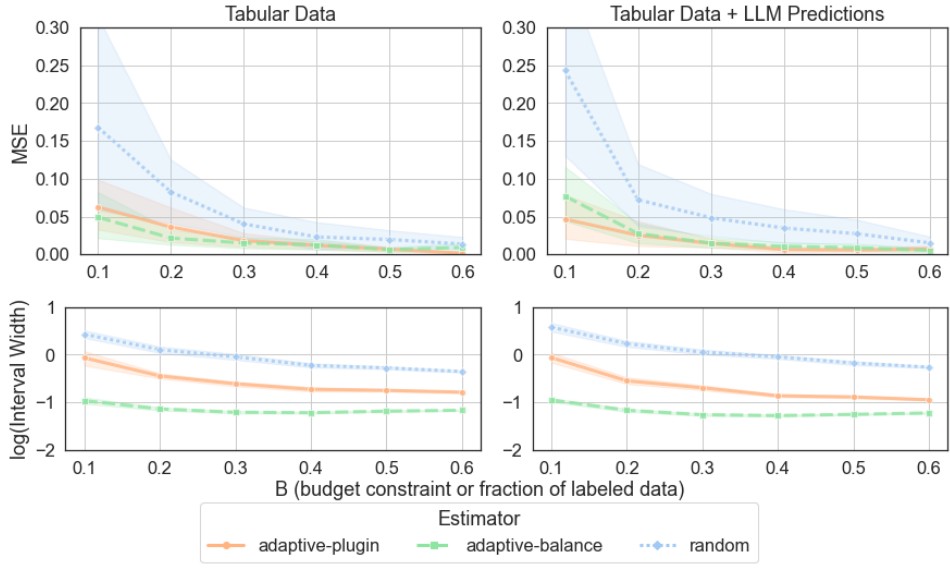

Figure 8: **Street outreach data with post-treatment summaries only.** Mean squared error and 95% confidence interval width averaged over 20 trials across budget percentages of the data. This plot makes use of tabular data and the best-performing random forest outcome model (left) and text-encoded outcomes using LLMs (right).

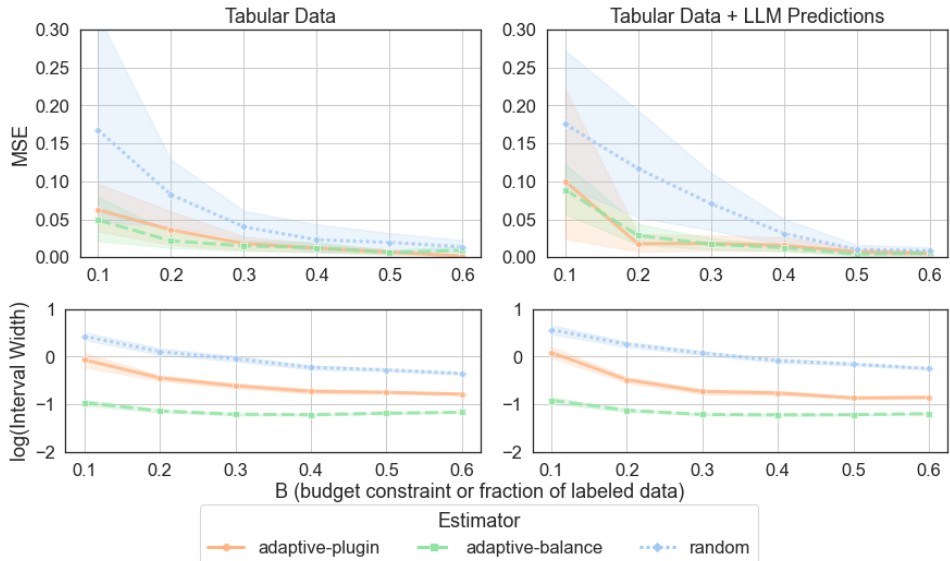

Figure 9: **Street outreach data with pre- and post-treatment summaries.** Mean squared error and 95% confidence interval width averaged over 20 trials across budget percentages of the data. This plot makes use of tabular data and the best-performing random forest outcome model (left) and text-encoded outcomes using LLMs (right).

In Figure 8 and Figure 9, we see that our results and analysis are preserved, and qualitatively similar. Our adaptive approach still shows improvements over uniform random sampling. The MSE is tripled when going from our adaptive estimators to random sampling in the tabular data. The MSE is five times higher when going from adaptive to random sampling in the setting where we have added LLM predictions using post-treatment summaries $\tilde{Y}$ only and it is nearly doubled when using both pre- and post-treatment summaries.

In this experimental setup, we find that tabular estimation with ground-truth validated codes overall performs comparably as using more advanced LLM estimation. In this setup, we use placement outcomes as the measure of interest, in part because it is (nearly) fully recorded in our dataset, and hence we can consider it as having access to the "ground-truth" outcome in our methodological setup. On the other hand, we also expect that casenotes are weakly informative of placement, as compared with other outcomes we might seek to extract from casenotes (but do not have the ground-truth for). Nonetheless, this validates the usefulness of the method, and we leave further empirical developments for future work.

### I.6 Budget Saved Plots

We compute the amount of budget saved due to our batch adaptive sampling approach. We find the sample size required to achieve the same confidence interval width with batch adaptive annotations using balancing weights (green) and RZ-plug-in (orange) compared to uniform random sampling.

### I.7 Active Learning Baselines

Active learning is not a strong baseline and we argue this on theoretical and empirical fronts. Active learning for regression can't improve statistical rates of convergence, while the doubly-robust AIPW estimator in causal inference can, so using AIPW is optimal. Additionally, using pool-based active learning algorithms in AIPW blows up variance due to near-deterministic annotation probabilities. Active learning models only target $\mu_z$, but the outcome model contributes $\frac{\sigma_z^2(x)}{e_z(x)\pi(z,x)}$ to the causal Avar, and our optimal annotation correctly balances the effect of all factors, but active learning only considers the first.

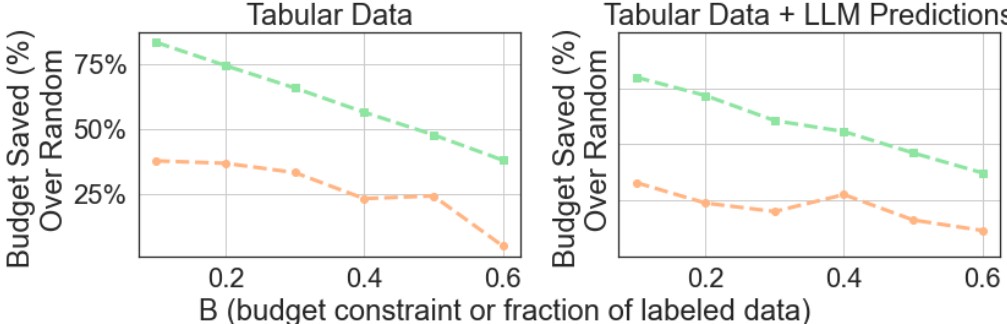

Figure 10: **RetailHero Data.** Budget saved due to batch adaptive annotation. The reduction in annotation sample size needed to achieve the same confidence interval width with batch adaptive annotation on tabular data (left) and on tabular data + complex embedded outcomes (right) compared to random sampling.

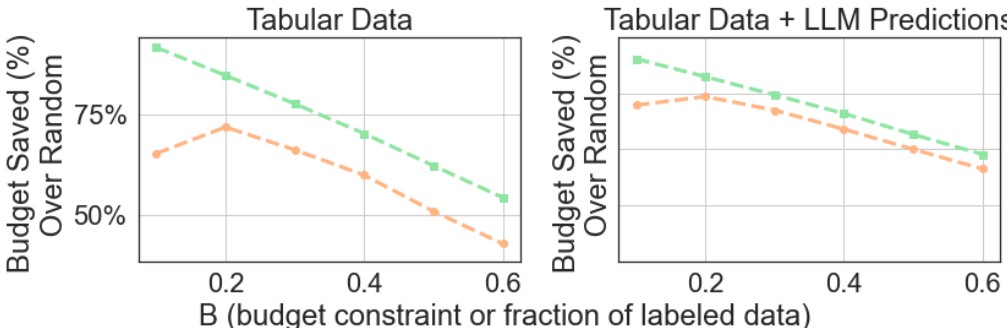

Figure 11: **Street Outreach Data.** Budget saved due to batch adaptive annotation. The reduction in annotation sample size needed to achieve the same confidence interval width with batch adaptive annotation on tabular data (left) and on tabular data + complex embedded outcomes (right) compared to random sampling.

In summary, active learning does something *completely different for prediction error, suboptimal for causal inference*.

Empirically, we run active learning algorithms to learn $\mu$ in AIPW and find that it *totally fails* for these reasons; if these objectives line up, it can do well, but in general, the prediction and causal error objectives are different.

**Theoretical comparison to active learning.** As a reminder, we optimize:

$$AVar_{ATE} = Var[CATE(X)] + \sum_{z \in \{0,1\}} E[\frac{\sigma_z^2(X)}{e_z(X)\pi(z,X)}]$$

(The first term is the variance of $CATE = E[Y(1) - Y(0)|X]$; it is never observed.)

To go more in detail on our experiments 1) we compare to theoretical results in batch *pool-based active learning*, Chaudhuri et al. [7] and Gentile et al. [22] (henceforth GWZ), which show that active learning doesn't improve convergence rates for regression, only multiplicative constants. Instead, the AIPW estimator is optimal for causal estimation: if the outcome and propensity scores can only achieve $n^{-1/4}$ convergence, the AIPW estimator is $O(n^{-1/2})$-rate convergent, so AIPW can speed up outcome model convergence rates. Therefore using the AIPW estimator is best, and random sampling + AIPW is a stronger baseline than active learning.

To emphasize the different objectives, consider a simple example with two regions:

- Region 1 (Poor Overlap), $X > 0$: Propensity score $e(X) = 0.01$; outcome noise $\sigma_1(X), \sigma_0(X)$=1.

- Region 2 (High Prediction Uncertainty), $X < 0$: Propensity score $e(X) = 0.5$; outcome noise $\sigma_1(X), \sigma_0(X) = 10$ and the outcome model is complex.

Our method compares the ATE variance contribution in either region:

- Region 1: $\frac{\sqrt{1}}{0.01} = 100$

- Region 2: $\frac{\sqrt{100}}{0.5} = 10$

and samples in Region 1, where the causal variance is five times higher. Uncertainty-based active learning samples in Region 2, to the detriment of causal variance.

**Active Learning Empirical Evaluations.** We evaluate our method against 2-3 active learning baselines for each experiment from two popular and well-established python packages (scikit-activeML and modAL). Different active learning algorithms are appropriate for different outcome models, so we choose the sampling strategy based on our modeling task, and we use pool-based active learning matching our two-batch approach. (Note our approach is *model-agnostic*, while active learning methods are not). For the classification tasks on our two real-world datasets (RetailHero/Street Outreach), we use UncertaintySampling with margin sampling and least confident sampling as query strategies, which both choose x with highest uncertainty measure based on classification probabilities $P(\hat{Y} = 1 \mid x)$ [40]. For the regression tasks, we use Expected Model Variance Reduction [12], Expected Model Change Maximization [6], and Improved Greedy Sampling [51]; these choose $x$ that maximizes greatest future variance reduction, maximally change the current model via the loss gradient, and diversity in feature and output space, respectively.

We run each approach over 50 trials and take the average MSE. Across the board, we see that our approach does better than the popular active learning strategies that are not optimized for causal estimation.

**Result Tables**

| Estimator | 0.1 | 0.2 | 0.3 | 0.4 | 0.5 | 0.6 | 0.7 | 0.8 | 0.9 |
|---|---|---|---|---|---|---|---|---|---|
| active-evar | **0.313** | 17.3 | 85.1 | 579 | 1.31e+03 | 3.87e+03 | 1.27e+04 | 5.03e+04 | 8.93e+05 |
| active-greedy | 6.13 | 79.9 | 369 | 852 | 1.99e+03 | 5.06e+03 | 1.33e+04 | 5.09e+04 | 2.95e+05 |
| active-mvar | 10.6 | 94.3 | 314 | 883 | 2.17e+03 | 5.70e+03 | 1.21e+04 | 3.87e+04 | 2.99e+05 |
| adaptive-balance | 0.471 | **0.227** | **0.276** | 0.236 | **0.265** | **0.246** | **0.198** | **0.176** | **0.203** |
| adaptive-plugin | 1.7 | 1.17 | 0.831 | **0.196** | 0.83 | 0.449 | 0.507 | 0.93 | 0.481 |
| random | 8.99 | 4.56 | 2.19 | 1.54 | 1.7 | 1.61 | 1.46 | 0.956 | 0.987 |

Table 3: Averaged MSEs for Synthetic Data.

| Estimator | 0.1 | 0.2 | 0.3 | 0.4 | 0.5 | 0.6 | 0.7 | 0.8 | 0.9 |
|---|---|---|---|---|---|---|---|---|---|
| active-margin | 3.53e+03 | 0.047 | 0.087 | 12.5 | 8.38e+03 | 2.25e+06 | 1.49e+06 | 6.53e+05 | 1.43e+07 |
| active-uncertain | 16.1 | 38.9 | 70.4 | 75.9 | 115 | 112 | 168 | 250 | 402 |
| adaptive-balance | **0.004** | 0.002 | 0.002 | **0.001** | **0.001** | 0.001 | **0** | **0** | **0** |
| adaptive-plugin | **0.004** | **0.001** | **0.001** | **0.001** | **0.001** | **0** | **0** | **0** | **0** |
| random | 0.027 | 0.012 | 0.009 | 0.006 | 0.005 | 0.003 | 0.001 | 0.001 | **0** |

Table 4: Averaged MSEs for RetailHero Data.

| Estimator | 0.1 | 0.2 | 0.3 | 0.4 | 0.5 | 0.6 | 0.7 | 0.8 | 0.9 |
|---|---|---|---|---|---|---|---|---|---|
| active-margin | **0.009** | 28.5 | 4.47 | 0.501 | 0.449 | 0.044 | 0.099 | 0.412 | 0.209 |
| active-uncertain | 0.017 | **0.009** | 0.018 | 0.008 | 0.017 | 0.018 | 0.025 | 0.023 | 0.024 |
| adaptive-balance | 0.046 | 0.031 | **0.013** | **0.006** | **0.005** | **0.003** | **0.004** | **0.003** | 0.002 |
| adaptive-plugin | 0.045 | 0.025 | 0.027 | 0.012 | 0.006 | 0.004 | **0.004** | 0.006 | **0.001** |
| random | 0.113 | 0.061 | 0.037 | 0.045 | 0.014 | 0.012 | 0.011 | **0.003** | **0.001** |

Table 5: Averaged MSEs for Street Outreach Data.

Gentile et al. [22] chooses a point x maximizing a diversity measure, D(x,S) that quantifies model uncertainty and is directly influenced by the observation noise, $\sigma_z^2(X)$. For general function approximation, they introduce a maximal disagreement measure over the regression function class $\mathcal{F}$ $\sup_{f,g \in \mathcal{F}} \frac{(f(x)-g(x))^2}{\sum_{z \in S}(f(z)-g(z))^2+1}$, where $S$ is the set of already sampled points. If $\sigma^2(x)$ is large for some $x$, their disagreement measure is also large. Their diversity measure finds points where it is possible for two functions, $f, g$, to have similar predictions on the already-labeled data S (a small denominator) but different predictions for a new point $x$ (a large numerator). When observation noise $\sigma^2(x)$ is larger, many different functions can be considered "plausible" fits and can agree on S but disagree elsewhere, leading to a high diversity score. In contrast, low noise tightly constrains all plausible functions, resulting in low disagreement.

## I.8 LLM Prompts

**Prompt 1 (Retail Hero):**

You are a user who used a website for online purchases in the past one year and want to share your background and experience with the purchases on social media.
*Attributes:*
The following are attributes that you have, along with their descriptions.
$\{features\}$
# Personality Traits The following dictionary describes your personality with levels (High or Low) of the Big Five personality traits.
$\{traits\}$
*Your Instructions:*
Write a social media post in first-person, accurately describing the information provided.
Write this post in the tone and style of someone with the given personality traits, without simply listing them.
Only return the post that you can broadcast on social media and nothing more.
—

$\{post\}$
–_

**Prompt 2 (Street Outreach Casenote Summaries) :**

Objective: Your task is to summarize a trajectory of case notes of a client in street home-lessness outreach, focusing on client interactions, the challenges they are facing, goals they are working towards, and progress towards housing placement. These are all from the same client. This summary is designed to help caseworkers and organizations assess client history at a glance, remind of prior personal information and important challenges mentioned (like veteran status or other information that is relevant for eligibility for housing, medical issues, and status of their support network), allocate resources effectively, and improve support for individuals experiencing chronic homelessness.

Context: $\{task\_context\}$

The summary should be a concise overview of the client's situation, highlighting key points from the case notes. It should not include any personal opinions or assumptions about the client's future or potential outcomes. The goal is to provide a clear and informative summary that can be used by caseworkers and organizations to better understand the client's history and current status.

Here are the case notes for batch $\{batch\_num\}$ of $\{total\_batches\}$:

— START NOTES —

$\{notes\}$

— END NOTES —

Based *only* on the notes provided above for this batch, generate a comprehensive summary focusing on key events, decisions, and progress during this specific period. The target length is approximately $\{target\_length\}$ words. Ensure the summary strictly reflects the content of these notes.

957

**Prompt 2 (Street Outreach Classification) :**

You are an expert analyst specializing in predicting long-term housing stability for individuals experiencing homelessness. Your task is to analyze client data, including demographic information, historical interactions, and case note summaries, to predict the **most stable housing placement level** the client is likely to achieve and maintain over the **next two years**.

**Input Data:**

You will be provided with the following information for each client:

**Prediction Task:**

Based *only* on the provided attributes and the case notes summary, predict the single most stable housing placement level the client is likely to maintain over the next two years.

**Housing Placement Levels (Prediction Output):**

Your prediction must be an integer between 0 and 3:

- **0**: No stable placement (remains on the street or in emergency shelters).

- **1**: Transitional Housing (temporary placement with support, aiming for longer-term housing).

- **2**: Rapid Re-housing (time-limited rental assistance and services).

- **3**: Permanent Supportive Housing (long-term housing with ongoing support services).

**Reasoning Guidance (Internal Thought Process - Do Not Output This):**

- Consider factors that promote stability: housing application progress, possession of documents, benefit acquisition, engagement with services (unless contacts are excessive without progress), prior successful placements (even if temporary), positive recent developments in the case notes.

- Consider factors that hinder stability: chronic homelessness indicators, frequent service refusals, mental health crises (Removal958), lack of documents/income, lack of prior placements, patterns of instability noted in the summary.

- Weigh the structured data against the nuances presented in the case note summary. The summary provides vital context.

**Client Information:**

**Prediction:**

Provide *only* the predicted number (0, 1, 2, or 3) as the output. Do not include any other text, explanation, or formatting.

**Examples:** $\{examples\}$

