# OpenReview forum: "Batch-Adaptive Annotations for Causal Inference with Complex-Embedded Outcomes"
_NeurIPS.cc/2025/Workshop/Reliable_ML — NeurIPS 2025 - Reliable ML Workshop_

### Official Review · Reviewer_mUQs · 2025-09-19
**This paper addresses a significant practical challenge in causal inference involving three key complexities: (1) treatment-dependent outcomes, (2) missing outcome data, and (3) outcomes that represent complex transformations of the underlying true outcomes rather than direct measurements.**

**Rating:** 7
**Confidence:** 3

**Review:**

This problem is significantly important in the practical scenario, where data collection is not ideal. So this is a significant problem. Furthermore, the authors try to tackle all three aspects simultaneously,

1) output depends on the treatment,

2) some of the outputs might be missing and

3) The output might not be direct but just a complex function of the true output.

Strengths:

They establish their claims both theoretically, and I think the proofs are okay. There are also enough empirical results as well do support their claim.

Weaknesses:

However, their methodology seems to be very basic. They just implemented a two-batch adaptive design and compared it with a trivial baseline. I understand the complexity of the problem too much to implement a parallel exploit and explore a kind of algorithm. However, it is worth a try. Instead of using one batch totally to explore and then exploit the learned information on the second batch, we can try something to update a model while exploiting, similar to bandit algorithms.

There is also a minor suggestion: It was difficult to find out what ATE and AIPW stand for. I am not sure whether they are defined at all.

However, I am inclined towards the acceptance of the paper for the practical significance of the problem they are analyzing. Due to the time limit, I could not go through the whole appendix.

---

### Official Review · Reviewer_CMqt · 2025-09-19
**Review for "back arrowBack to Reviewers Console Batch-Adaptive Annotations for Causal Inference with Complex-Embedded Outcomes"**

**Rating:** 6
**Confidence:** 2

**Review:**

The paper studies an interesting and practical problem arising in causal inference, namely how to estimate treatment effects when outcome data is costly to obtain, for example when a non-profit is trying to assess the effects of a specific policy intervention to tackle homelessness but it is infeasible to go and ask every single person affected whether or not they are still homeless. The problem is very well-motivated and the authors propose a two-stage, batch-adaptive annotation strategy to decide which data points to label in order to reduce variance in treatment effect estimates. In the first stage, a random sample is selected, as one would expect, to estimate the nuisance functions for the asymptotic variance on the fully observed data. Then, in the second stage, these estimates are used to estimate the optimal labeling probabilities. The results from both simulated and applied datasets seem solid and demonstrate meaningful efficiency gains over random labeling.

Strengths:
* The authors' formal guarantees for their estimator are quite nice, showing that it achieves the optimal asymptotic variance, and they validate the approach across multiple datasets, including one based on sensitive, real-world service data.

* The paper also bridges methods from causal inference, adaptive experimental design, and machine learning-based outcome modeling.

Weaknesses:
* The method assumes that expert annotations reveal ground truth, though in practice annotators could disagree or have inherent biases.

Overall, the paper presents a well-motivated and technically solid method for an important problem and the authors' guarantees of the optimal asymptotic variance is a very nice contribution.

---

### Official Review · Reviewer_g6dN · 2025-09-21

**Rating:** 9
**Confidence:** 4

**Review:**

### Summary

This paper proposes an algorithm for Average Treatment Effect (ATE) estimation,
$
\mathbb{E}[Y(1)-Y(0)],
$
when outcomes are missing but can be retrieved at a cost. Given covariates $X\$ and a binary treatment $Z\in\{0,1\}$, the question is: how should we choose which units to annotate (i.e., measure the true outcome $Y$) under an annotation budget $B$?

The method assumes unconfoundedness and overlap, and uses the doubly-robust AIPW estimator for the ATE. The objective is to choose an annotation probability $\pi(z,x)$ to minimize the asymptotic variance of AIPW. This variance depends on nuisance quantities: the propensity score $e_z(x)=P(Z=z\mid X=x)$ and the conditional outcome variance $\sigma_z^2(x)=\mathrm{Var}(Y\mid Z=z,X=x)$.

The algorithm is two-batch. In Batch 1, a small, uniformly sampled set is annotated to estimate the nuisance functions. These estimates are used to compute an (approximately) optimal $\pi(z,x)$ for Batch 2, which is then annotated according to $\pi$. Finally, both batches are combined with cross-fitting to compute the ATE using AIPW.

The framework extends to “complex embedded outcomes,” where every unit has an available proxy outcome $\tilde Y$ (e.g., text or images) but the true $Y$ is costly to obtain. The proxy can improve outcome modeling $\mu_z(x)=\mathbb{E}[Y\mid Z=z,X=x]$, while annotation design still depends on $(X,Z)$. Experiments on simulated and real data support the approach.

---

### Strengths

- Clear presentation of the problem, method, and theory.
- Builds on the standard AIPW estimator; the use of $\pi$ is intuitive and easy to integrate.
- Addresses a practical and timely problem: how to spend limited labeling budget to improve causal estimation.
- Real-world application (social services) highlights practical impact.

---

### Weaknesses

- The experiments sometimes use LLM-derived features to find nuance parameters. As LLMs are black boxes, there is no consistency guarantee for the nuisance estimation.
- Unconfoundedness and overlap are standard assumptions but often violated in practice. The paper would benefit from justification (domain knowledge or diagnostics) for why these assumptions are plausible in the datasets used.

---

### Suggestions

1) Assumptions and diagnostics: Provide simple diagnostics or domain arguments for unconfoundedness and overlap (e.g., propensity score histograms, minimum-overlap thresholds).

2) Further study the use of LLMs in the nuance parameter estimation subroutine. What are the implications of using this tool?

---

### Overall

I believe the authors tackle a novel problem in causal inference literature: estimating ATE from missing data. Their application to real world data from social services highlights the practical significance of the task.